evolution, neuroscience, cognition

encephalization, trade-offs, experimental evolution

**Authors for correspondence:**
Ewelina Knapska
e-mail: e.knapska@nencki.edu.pl
Marek Konarzewski
e-mail: marekk@uwb.edu.pl

# Brain size, gut size and cognitive abilities: the energy trade-offs tested in artificial selection experiment

Anna Goncerzewicz[1], Tomasz Górkiewicz[1], Jakub M. Dzik[1], Joanna Jędrzejewska-Szmek[1], Ewelina Knapska[1] and Marek Konarzewski[2]

[1]Nencki Institute of Experimental Biology, 02-093 Warsaw, Poland
[2]Faculty of Biology, University of Białystok, Ciołkowskiego 1 J, 15-245 Białystok, Poland

EK, 0000-0001-9319-2176; MK, 0000-0001-7428-6521

The enlarged brains of homeotherms bring behavioural advantages, but also incur high energy expenditures. The 'expensive brain' (EB) hypothesis posits that the energetic costs of the enlarged brain and the resulting increased cognitive abilities (CA) were met by either increased energy turn-over or reduced allocation to other expensive organs, such as the gut. We tested the EB hypothesis by analysing correlated responses to selection in an experimental evolution model system, which comprises line types of laboratory mice selected for high or low basal metabolic rate (BMR), maximum ($VO_{2max}$) metabolic rates and random-bred (unselected) lines. The traits are implicated in the evolution of homeothermy, having been pre-requisites for the encephalization and exceptional CA of mammals, including humans. High-BMR mice had bigger guts, but not brains, than mice of other line types. Yet, they were superior in the cognitive tasks carried out in both reward and avoidance learning contexts and had higher neuronal plasticity (indexed as the long-term potentiation) than their counterparts. Our data indicate that the evolutionary increase of CA in mammals was initially associated with increased BMR and brain plasticity. It was also fuelled by an enlarged gut, which was not traded off for brain size.

## 1. Introduction

Many studies suggest that variation in brain size is ecologically adaptive and maintained by selective trade-offs (e.g. [1,2]). In particular, humans stand out among other mammals by their extremely large brains, a trait that also results in higher energy expenditures [3]. How were the energetic costs of these enlarged brains (encephalization) overcome? What sort of energetic, anatomic and physiological trade-offs and/or inherent positive associations were involved? The 'expensive brain' (EB) hypothesis posits that the energetic costs of increased brain size were met in either of two partially complementary ways [4]. The first one proposes that encephalization was primarily possible thanks to 'financing' metabolic costs of brain maintenance by reducing the size of energetically demanding gut parts [5]. Such reduction was in turn possible by increased cognitive abilities (CA) that allowed for more efficient foraging for food of better quality. The second of suggested evolutionary pathways does not necessarily invoke the brain–gut trade off, but rather points to the overall increase of energy intake fueling the progress of encephalization [3].

The EB scenarios in essence refer to physiological mechanisms and are therefore difficult to test, because of the lack of palaeontological record. Their evolutionary plausibility can thus only be tested if they reflect more general evolutionary principles applicable to extant animals characterized by positive association between enlarged brains and enhanced CA. To date, studies on the EB scenarios have predominantly used comparative methods, which

yielded mixed results. The existence of the brain–gut trade-off has been questioned in a thorough comparative analysis of brain size and internal organ mass in 100 mammalian species, including 23 primates [6]. Conversely, mammals manifest a positive association between brain size and basal metabolic rate (BMR) [7]—a measure of aerobic metabolism reflecting in large part energetic costs of maintenance of both, the gut [8] and the brain [9]. Furthermore, brain size in mammals is positively correlated with their maximum aerobic metabolism [10]—a strong predictor of their geographical range size [11] and therefore a suite of organismal capacities related to foraging and the high rate of reproduction that must ultimately be fueled by the gut. These results are also incompatible with the brain–gut trade-off.

A stronger test of the EB and its associations with CA is provided by artificial selection experiments because they allow for inferences about causal relationships [12]. The most pertinent animal model of this kind was developed by Kotrschal *et al.* [13], who demonstrated the brain–gut trade-off in guppies (*Poecilia reticulata*) artificially selected for relative brain size. However, life history and physiology of fish are far removed from that of homeotherms [14,15], therefore its relationship with selection on encephalization in mammals, for example, is questionable.

Here, for the first time, we tested the EB hypothesis in a mammalian model of experimental evolution. We used line types of laboratory mice subjected to artificial selection on high or low basal (BMR), or high maximum ($VO_{2max}$) metabolic rates [16–18]—traits widely accepted as pre-requisites for the evolution of homeothermy and large brain size [19,20]. Our experimental model allowed us to analyse not only trade-offs between anatomic and physiological traits, but also directly test directionality of associations between energy expenditures and the rate of learning by means of a battery of carefully controlled behavioural tests. In particular, we tested for (1) the existence of the brain–gut trade off and (2) positive associations between BMR or $VO_{2max}$ and CA and brain size.

The great majority of earlier studies related to the EB hypothesis were founded on the implicit assumption that any increase in brain size must provide some increase in function. Brain size (mass) alone, however, may be a poor proxy of CA [21]. More importantly, it does not provide any information on the neurophysiological basis of the putative increase of such abilities. To address this limitation here we analysed the among-line type differences in the activity-dependent synaptic plasticity of neurons [22,23], which gave us a meaningful insight into neuronal mechanisms underlying the observed variation in learning abilities.

## 2. Results

### (a) BMR and organ mass
Mean body mass did not differ among mice from the line types subjected to artificial selection on high (H-) or low (L-) (BMR), nor maximum ($VO_{2max}$) metabolic rate (peak metabolic rate, PMR) and random-bred (unselected, RB) lines (table 1). Conversely, H-BMR mice were characterized by conspicuously higher body mass-corrected BMR then mice of all other line types (table 1 and figure 1a; electronic supplementary material, figure S2A). Their metabolically expensive internal organs (liver, heart and kidneys) were also larger than in mice of

**Table 1.** ANCOVA results for BMR and organ masses.

| | line type | body mass |
|---|---|---|
| body mass | $F_{3,6} = 1.69$ | — |
| | $p = 0.27$ | |
| BMR | $F_{3,6} = 38.05$ | $F_{1,212} = 82.19$ |
| | $p < 0.001$ | $p < 0.001$ |
| brain | $F_{3,6} = 0.70$ | $F_{1,132} = 4.09$ |
| | $p = 0.58$ | $p = 0.04$ |
| liver | $F_{3,6} = 10.18$ | $F_{1,132} = 28.56$ |
| | $p = 0.009$ | $p < 0.001$ |
| heart | $F_{3,6} = 29.93$ | $F_{1,136} = 13.91$ |
| | $p < 0.001$ | $p < 0.001$ |
| kidneys | $F_{3,6} = 6.34$ | $F_{1,135} = 48.85$ |
| | $p = 0.03$ | $p < 0.001$ |

other line types, in particular the L-BMR mice (figure 1b–d; electronic supplementary material, figure S2B–D). Yet, their brains were not significantly larger, in particular with respect to their direct counterparts—the mice from the L-BMR line type (figure 1e). This lack of statistically significant difference between brain masses of the H-BMR and L-BMR mice was in agreement with their weak separation expressed against the between line type difference expected under genetic drift (electronic supplementary material, figure S1). Weak separation of brain mass of mice from both line types contrasted with sizable separation of other internal organs (electronic supplementary material, figure S1). Thus, both a direct comparison of all line types and an evaluation of H-BMR versus L-BMR difference with respect to the effect of genetic drift showed neither brain–gut trade-off nor positive genetic correlation between BMR and brain size.

### (b) Behavioural tests
To compare learning abilities we trained mice in IntelliCages, an automated system that allows for individual assessment of activity and learning of group-housed mice [24]. In an initial acclimatization phase, mice were able to access water in any of the four corners of the IntelliCage—each corner had two separate bottles with tap water that the mouse could choose between. During the place preference learning, water access for each mouse was restricted to one of the four corners. Next, in the reward-seeking discrimination learning, one of the bottles was filled with a reward—10% sucrose solution (figure 2a)—and the learning progress was scored as the number of nosepokes that opened access to the bottle with reward (correct responses).

In comparison to the previous 24 h of previous phase of the training, during the next 24 h, all mice increased the number of nosepokes to the bottle that now contained the reward. However, H-BMR mice accessed the reward more often than their L-BMR, PMR and RB counterparts (table 2 and figure 2b). Most importantly, a highly significant line-type × day interaction indicated that the H-BMR mice learned the rewarded response faster than the other animals ($F_{1,757} = 15.0$; $p < 0.001$ for the planned comparison of the slope

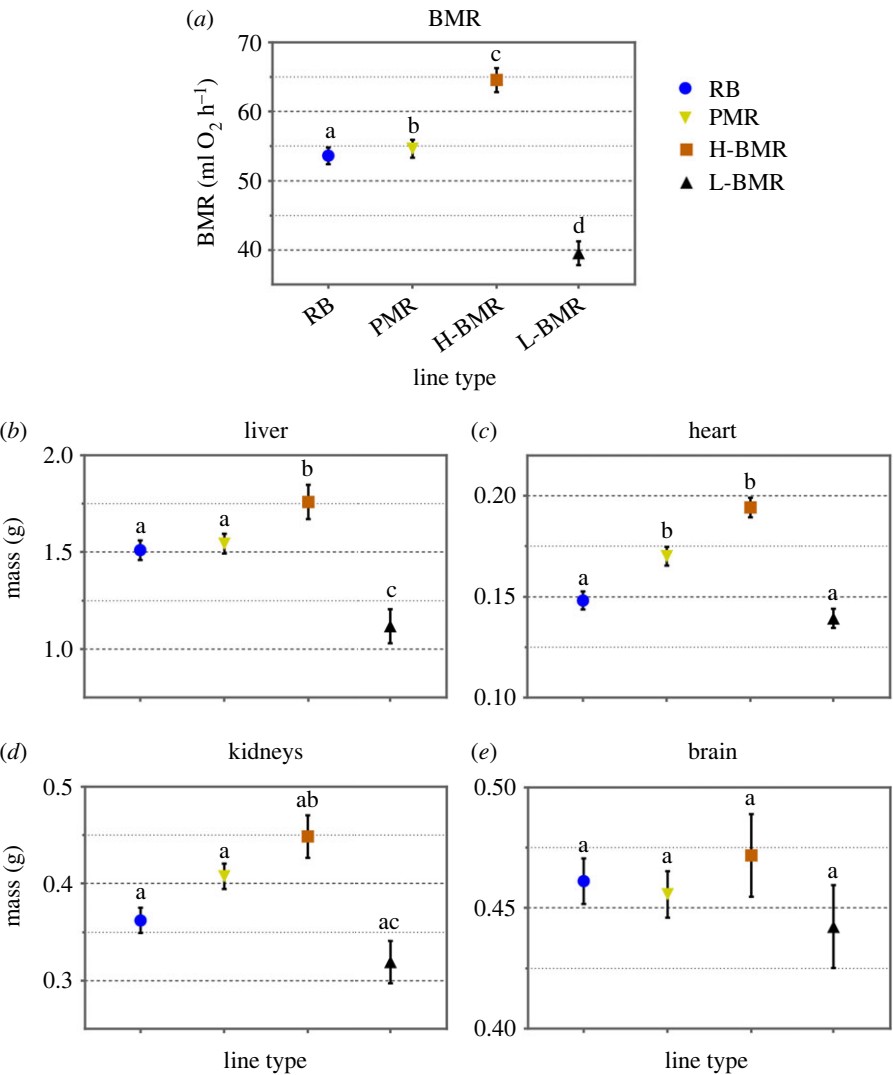

**Figure 1.** (*a*) Basal metabolic rate (BMR) in line types of mice used in the present study. (*b–d*) Masses of internal organs. (*e*) Brain mass. Values are body-mass-adjusted means with standard errors calculated from ANCOVA. Figure bars labelled with different letters differ significantly from each other at $p = 0.05$ (Bonferroni-adjusted for pairwise comparisons). In this and subsequent figures we use following abbreviations: RB, random-bred line type; H-BMR and L-BMR, mice of the line types selected for high or low basal metabolic rate (BMR), respectively; PMR, mice of the line type selected for peak metabolic rate (aka VO$_{2max}$). (Online version in colour.)

of change of the number of nosepokes in the H-BMR mice versus other line types).

To test whether the improved learning could be attributed to changes in thirst or taste discrimination, the number of licks from the bottles that contained sucrose solution was analysed. We did not observe any differences among the line types in the amount of sweetened water consumed (table 2). Further, because differences in general activity could potentially influence the obtained results, we compared the numbers of visits to all corners during the reward-seeking discrimination learning phase and the adaptation phase. The rate of visiting corners did not differ among the line types (table 2), excluding the possibility that the differences in learning could be explained by changes in general activity.

To exclude a possibility that the superior learning response of the H-BMR mice was solely limited to the reward-seeking context or higher motivation to perform a nosepoke response, we used another group of naive mice and carried out a study designed in the IntelliCage system as described above, but with the reward-seeking discrimination learning followed by an aversive cue discrimination task (figure 2*a*). In this additional task, mice learned to avoid an aversive cue provided by a water solution of 0.005 M quinine placed in one of the

IntelliCage corners. In the reward-seeking part of the trial, the H-BMR mice again accessed the reward more often than the mice of other line types (line-type × day interaction, $F_{3,306} = 7.13$; $p < 0.001$). By contrast, in the aversive discrimination learning task the number of nosepokes to the bottles now containing its solution decreased in the H-BMR mice and remained unchanged in other line types (figure 2*c*). This resulted in a significant line-type × day interaction (table 2), that was due to a reduction of the nosepokes in the H-BMR mice, as only in this line type the number of nosepokes significantly dropped ($F_{1,308} = 5.1$; $p = 0.02$ for the planned comparison of the slope of change of the number of nosepokes in the H-BMR mice versus other line types; figure 2*c*). Thus, the H-BMR mice learned to avoid aversive cue faster than the mice of other line types.

Finally, in yet another group of naive mice we investigated the differences in learning abilities among line types using a classic paradigm of contextual fear conditioning [25]. Following conditioning elicited by a mild electric foot shock applied in a novel context we measured extinction of freezing response to perceived threat (i.e. in the absence of the electric shock). The line type × time interaction was statistically significant ($F_{15,30} = 2.27$; $p = 0.03$), which reflected

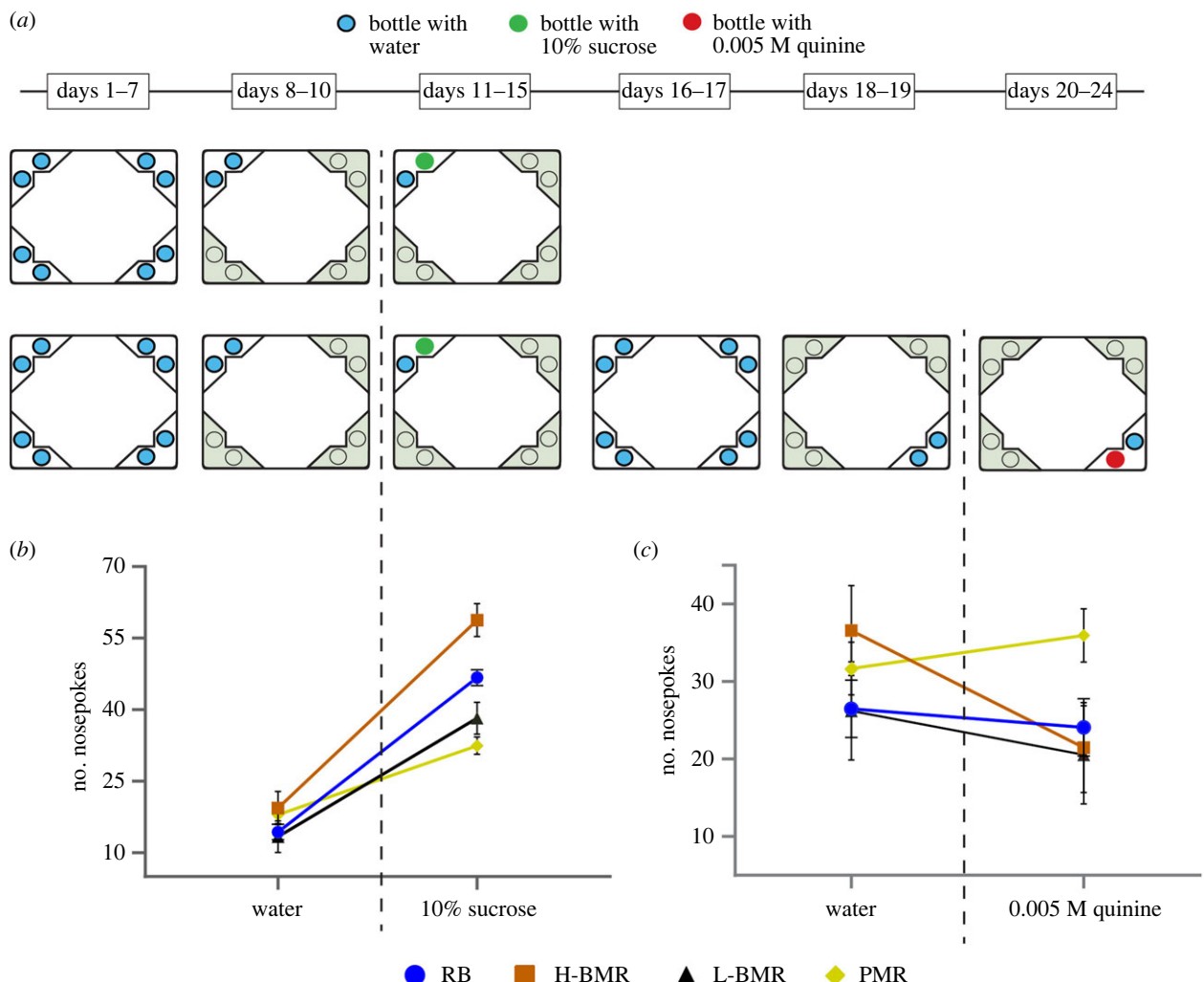

**Figure 2.** Scheme of the experiments in the IntelliCage. (*a*) In the reward-seeking discrimination learning task (upper raw of schemes) naive mice were subjected to experimental procedures that consisted of simple adaptation phase (days 1–4, not shown), nosepoke adaptation phase (days 5–7), place preference learning (days 8–10) and reward-seeking discrimination learning (reward: 10% sucrose solution, days 11–15). These phases were repeated in the aversive discrimination task (lower row of schemes), in which another group of naive mice was also subjected to additional phases: the nosepoke adaptation (days 16–17) and place preference learning to a different corner (days 18–19). Next, in days 20–24 mice were exposed to aversive discrimination learning procedure with a 0.005 M quinine solution. (*b*) Results of the reward-seeking discrimination learning: number of nosepoke responses giving access to the bottle that contained tap water in preference learning, then sweetened water (aligned by vertical dashed line with the timeline of experiment). Values are least-squares means (±s.e.) of nosepokes from the repeated measures mixed ANCOVA. Slopes of the lines depicting the H-BMR line type significantly differ from the slopes of the remaining lines at *p* = 0.05 (by *a priori* custom made contrast). (*c*) Least-squares means (±s.e.) as in (*b*), but of the number of incorrect nosepokes counted in an aversive cue discrimination learning task, in which we used water solution of 0.005 M quinine. (Online version in colour.)

**Table 2.** Repeated measures ANCOVA results for behavioural tests.

| | line type | Day | Period | Day × Period | line type × Period | line type × Day |
|---|---|---|---|---|---|---|
| correct nosepokes sucrose[a,b] | $F_{3,6} = 5.45$ | $F_{1,757} = 345.5$ | $F_{1,757} = 88.5$ | $F_{1,757} = 88.5$ | $F_{1,} = 1.1$ | $F_{3,757} = 19.8$ |
| | $p = 0.03$ | $p < 0.001$ | $p < 0.001$ | $p < 0.001$ | $p = 0.3$ | $p < 0.001$ |
| activity[b] | $F_{3,6} = 0.92$ | $F_{1,803} = 21.3$ | $F_{1,803} = 467.3$ | $F_{2,803} = 43.7$ | $F_{1,803} = 28.55$ | $F_{3,803} = 5.1$ |
| | $p = 0.48$ | $p < 0.001$ | $p < 0.001$ | $p < 0.001$ | $p < 0.001$ | $p = 0.002$ |
| licks[b,c] | $F_{3,6} = 0.36$ | $F_{1,713} = 204.3$ | $F_{1,713} = 259.0$ | $F_{2,713} = 32.3$ | $F_{1,713} = 5.3$ | $F_{3,713} = 5.0$ |
| | $p = 0.78$ | $p < 0.001$ | $p < 0.001$ | $p < 0.001$ | $p = 0.001$ | $p = 0.002$ |
| incorrect nosepokes quinine[a] | $F_{3,6} = 1.27$ | $F_{1,308} = 8.4$ | $F_{1,308} = 11.4$ | $F_{1,308} = 4.1$ | $F_{1,308} = 2.35$ | $F_{3,308} = 2.88$ |
| | $p = 0.36$ | $p = 0.004$ | $p < 0.001$ | $p = 0.04$ | $p = 0.07$ | $p = 0.04$ |

[a]The numbers of responses (i.e. nosepokes to the bottle with sucrose or quinine) were corrected for numbers of nosepokes to the bottle with tap water located in the same corner (used as a covariate, significant at *p* < 0.001).
[b]In this analysis the effect of a batch of animals simultaneously subjected to behavioural test was significant as a fixed factor (*p* < 0.01).
[c]The numbers of licks of the bottle with sucrose were corrected for numbers of licks to the bottle with tap water located in the same corner (used as a covariate, significant at *p* < 0.001).

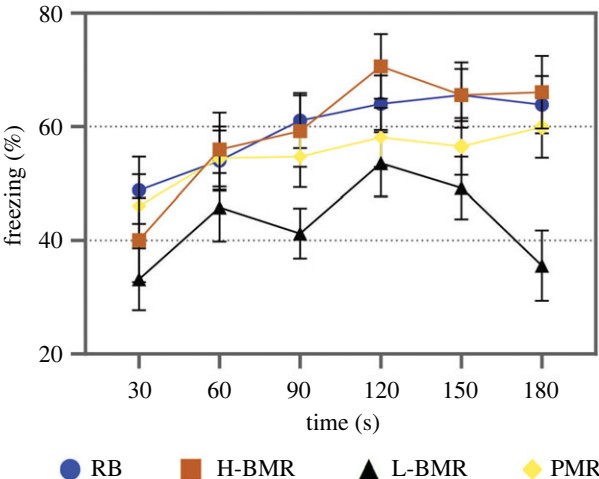

**Figure 3.** Changes in fear response. Freezing (immobility) was measured as [%] of total observations in 30 s intervals in high (H-BMR) and low (L-BMR) basal metabolic rate, peak metabolic rate (PMR; aka VO$_{2max}$) and random-bred (RB) line types of mice (means ± s.d.). Slope of the line depicting the L-BMR line type significantly differs from the slopes of the remaining lines at $p = 0.05$ (by *a priori* custom made contrast). (Online version in colour.)

the heterogeneity of the dynamics of fear extinction in the studied line types, accounted for the L-BMR mice losing fear response much faster than other line types (figure 3).

## (c) Neuronal plasticity

We compared the slope of the long-term potentiation (LTP) in the H-BMR mice, L-BMR mice and the animals from one of the RB (non-selected) lines as the reference group. In line with the behavioural results, the H-BMR mice manifested significantly increased neuronal plasticity as compared with the mice of the other line types ($F_{2,24} = 18.4$ $p < 0.001$; figure 4). Furthermore, the difference in LTP between the H-BMR and L-BMR mice was far larger than that expected under genetic drift (electronic supplementary material, figure S1), which suggests the existence of a positive genetic correlation between BMR and LTP.

## 3. Discussion

According to the EB hypothesis, the costs of increased brain size and CA can be satisfied by (i) reallocation of resources toward brain growth and maintenance from other energetically expensive organs, as proposed by Aiello & Wheeler [5]; or by (ii) increasing total energy intake, which may allow to cover the costs of CA without the need for reduction of other structures and functions, including digestive abilities [4]. Here, for the first time we comprehensively tested (i) and (ii) in a mammalian experimental evolution model. Our results do not support the existence of the brain–gut trade off envisaged in (i). It is important to note, however, that throughout our experiment mice were fed the same diet, so the partial tenet of the brain–gut trade off—compensation of the reduced gut by increased food quality [26] could not be tested. Yet, at least in non-mammalian animal models, the brain–gut trade-off is likely to occur even without a shift in quality of consumed food, as demonstrated by [13]. Also, as we demonstrated elsewhere [27] H-BMR mice possess a considerable digestive safety margins, which would have left them an ample potential for gut size reduction envisaged by the brain–gut trade-off.

An increase in energy intake is the hallmark of the evolution of endothermy [28], particularly linked with the need to fuel reproduction [29]. H-BMR mice are characterized by both increased energy intake and reproductive allocation [30] and increased mass of the gut (table 1). This points to (ii) and suggests that the selection for enhanced CA does not need to involve brain–gut trade-off as an initial step toward the evolution of enhanced CA. On the contrary, the H-BMR mice having larger guts, but not brains, performed better in cognitively demanding tasks than their L-BMR counterparts and mice selected for maximum aerobic metabolism (PMR line type). We compared behaviour of mice of all line types when highly rewarding 10% sucrose solution or unpleasant taste of 0.005 M quinine solution appeared in the IntelliCage system. The H-BMR animals increased number of nosepokes giving access to the sucrose solution and decreased nosepokes leading to the unpleasant bitter taste to a higher degree than the mice of other line types. Conversely, contextual fear conditioning test, in which animals learned the association between the novel cage (new context) and unpleasant foot shocks revealed that L-BMR mice performed worse than the other animals (figure 3). Overall, the results of behavioural tests point to the positive association of CA with the evolution of BMR, rather than maximum aerobic metabolism (VO$_{2max}$, selected for in the PMR line type), which has also been implicated in the evolution of homeothermy and large brain size [10,19,20].

The above-discussed differences among line types demonstrated in behavioural tasks beg the question of the underlying neuronal mechanism. We identified such a mechanism in the context of the EB hypothesis. Both, aversive and positive learning, such as in the tests we performed in IntelliCages, involve hippocampus [31]. Since the H-BMR mice performed best in the IntelliCage tests (figure 2b,c), while L-BMR mice seem particularly inferior with respect to fear conditioning (figure 3), we focused on identification of the relevant neuronal mechanisms differentiating mice from those two line types, using the RB mice as the reference group (figure 4). We tested hippocampal neuronal plasticity using a well-established LTP model [32]. LTP is an increase in signal transmission between neurons caused by strengthening of synapses by recent patterns of activity. LTP is considered one of the major cellular mechanisms of learning and memory formation [33]. Furthermore, excitatory synaptic transmission requires ATP-dependent phosphorylation of AMPA receptors [34] and therefore should be positively associated with the rate of aerobic metabolism. Our data show increased potentiation in the H-BMR mice and downregulation in the L-BMR animals, when compared to the RB mice, which suggests upregulation of physiological processes intrinsic to learning in the H-BMR line type.

In conclusion, we revealed the likely directionality of the evolutionary relationships between energy expenditures, brain, gut and CA in a mammalian model of experimental evolution. Our results point to an evolutionary scenario that would involve initial selection for increased overall energy intake, which would necessitate its positive genetic correlations with increased gut size and BMR [35]. This selection may have involved an initial increase in neuronal plasticity, if brains built of more plastic neurons were metabolically cheaper and cognitively more effective than the ones built of larger number of neurons of lower plasticity [36]. Such smarter (but not necessarily bigger) brains allowed for

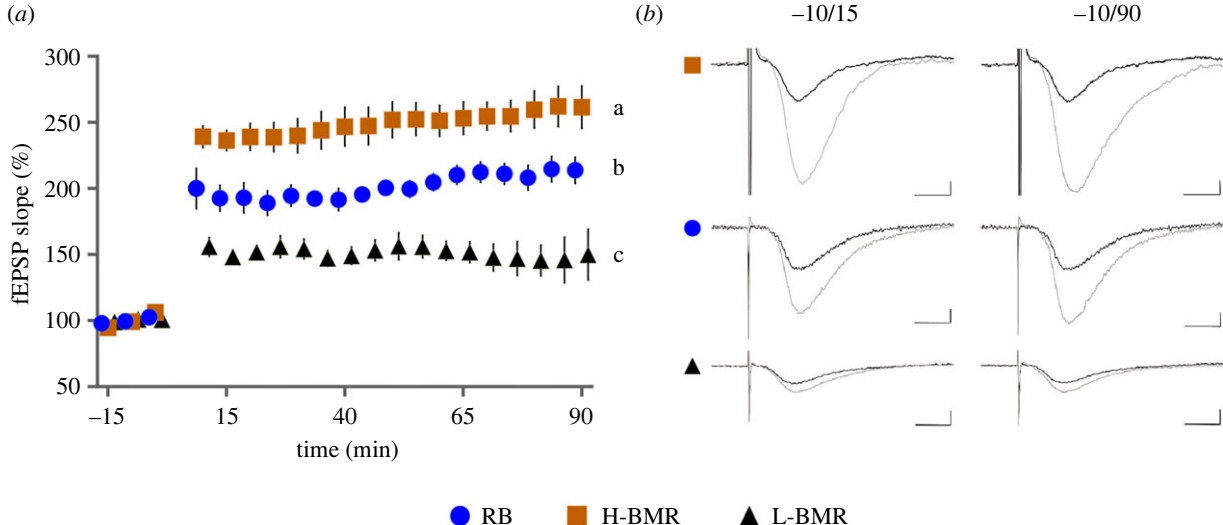

**Figure 4.** LTP recorded at the Shaffer correlates in the hippocampus. (*a*) The time course of maximal EPSP slopes was normalized to baseline in the CA1 region of the hippocampus. Long-term potentiation was induced by high-frequency stimulation (HFS; $3 \times 100$ Hz) of the Schaffer collaterals in slices from the H-BMR (orange squares, $n = 10$), L-BMR (black triangles, $n = 8$) and the RB mouse line (blue circles, $n = 7$). The slopes labelled with different letters differed from each other at $p = 0.05$ by the Tukey *post hoc* test. (*b*) Representative traces of fEPSP 10 min before (black) and 15 and 90 min after (grey) the induction of LTP are shown. Scale bars $= 2$ mV and 5 ms. Values are least-squares means ($\pm$s.e.) from the repeated measures mixed ANOVA. (Online version in colour.)

foraging on better-quality food, for example. Subsequently other trade-offs (such as gut reduction) may have occurred in some lineages, such as proto-human apes, allowing for further brain size increases [3].

# 4. Material and methods

## (a) Animals

We used 3–4-month old female mice from two concurrent selection experiments carried out at the Faculty of Biology, University of Bia-łystok. In the first experiment, we maintain two line types of mice divergently selected for high (H-BMR) or low (L-BMR) body-mass-corrected BMR, quantified according to the measurement procedure outlined below. The resulting divergence between those two non-replicated line types is sufficiently large to be confidently attributed to the applied selection, rather than to genetic drift (electronic supplementary material). We also used female mice from the second selection experiment, in which we established eight genetically isolated Swiss-Webster laboratory mouse lines. In four of the lines, forming the peak metabolic rate (PMR) line type, mice were selected for $VO_{2max}$ quantified as the highest body-mass-corrected oxygen consumption averaged over 2 min of a 5 min swim in a 25°C water [17,18]. The other four lines form the RB, control line type [17,18]. For further information on animal maintenance and the number of animals used in specific trials and analyses see electronic supplementary material.

## (b) Measurements of basal metabolic rate

We used an open respirometry system [16], ensuring high repeatability of measurements of individual BMR [37]. For detailed description of the system see electronic supplementary material.

## (c) Measurements of cognitive abilities

Following BMR measurements the mice were tested in an automated learning apparatus, an IntelliCage system, from TSE Systems, Germany [38,39]. We employed long-running automated behavioural tests carried out in a home cage to measure voluntary behaviors of mice, which were not water- and food-deprived. The IntelliCage consists of a large standard cage 20.5 cm high,

40 cm × 58 cm at the top and 55 cm × 37.5 cm at the base. The cage is equipped with four operant learning chambers fitted into the corners of the housing cage. Access into the chamber is only possible through a tube with a built-in transponder codes reader (antenna) that restricts access to the learning chamber to only a single mouse at a time. Each corner, equipped with proximity sensor, contains two openings permitting access to drinking bottles. An automatically operated door controls access to liquid. Poking a nose into the openings (nosepoke response) activates an infra-red beam-break response detector. Each visit to the operant chamber, as well as each nosepoke and the amount of water consumed (number and duration of licks) is recorded for each individual animal. The cage control unit permits the access to particular bottles according to schedules individually pre-programmed for each mouse. The cage is equipped with a sleeping shelter in the center, with a feeder placed on its top providing food *ad libitum*. Except for the technical breaks and cage exchange (once a week), the mice were not disturbed.

A week before the experiment the mice were sedated with isoflurane and injected with a glass-covered microtransponder (11.5 mm length, 2.2 mm diameter; DataMars) with a unique code recognized by sensors installed in the learning chambers [40]. After the transponder implantation procedure, subjects were moved from the housing facilities to the experimental rooms. The animals were then transferred to three IntelliCage systems, each housing 10–12 mice randomly drawn from the stocks of their parental lines. We housed individuals from the same line type to minimize possible effect of social context [40]. The number of mice living in the cage was adjusted so as to minimize competition for the access to the bottles [24].

Mice housed in each of the IntelliCages were maintained in a 12 : 12 light schedule (same as the maintenance conditions in their home animal facility) and subjected to an appetitive learning task for 15 day protocol divided into four phases: simple adaptation, nosepoke adaptation and place preference learning and reward-seeking discrimination learning (figure 2*a*). During the simple adaptation phase (days 1–4), all doors in the learning chambers remained open and access to water was unrestricted. During the nosepoke adaptation phase (days 5–7), all doors were closed and opened only when an animal poked its nose (nosepoke response) into one of the two openings placed inside learning chambers. When an animal removed the snout from the opening, the door closed automatically. During the simple

adaptation and nosepoke adaptation phase each of eight bottles contained tap water (days 1–7, figure 2a). During the place preference learning phase (days 8–10) access to the drinking bottles was restricted to only one of the IntelliCage learning chambers for each mouse.

The corner with water access was assigned randomly to no more than three mice. Such procedure minimized social modulation of learning [40]. During reward-seeking discrimination learning phase, tap water in one bottle in the corner was replaced by 10% sucrose solution, which is strongly preferred by mice (days 11–15). Animals had a choice between nosepoking (operant response) to the bottle containing tap water or to the bottle containing a reward (sweetened water) placed in the same conditioning corner. They had to remember location of the reward to perform the correct response depicted in figure 2b.

In aversive learning task we subjected a group of naive mice to the above-described reward-seeking discrimination learning procedure extended by three additional phases (figure 2a). In the first one, mice had access to water in all four corners for 2 days (days 16–17). During the next 2-day phase, mice had access to bottles only in one of the corners, which was different from the corner with the reward in the reward-seeking discrimination training. In the third phase, lasting 5 days, the bottle preferred during the previous two days was replaced with a bottle containing 0.005 M quinine solution evoking aversive, bitter taste perception in mice of all studied line types. Changes in the number of nosepokes to the bottles containing quinine (i.e. incorrect responses) recorded during the first critical 24 h are depicted in figure 2c.

The number of visits, nosepokes and tube licks was recorded automatically by the computer-controlled IntelliCage system in 12 h time intervals. All raw data were then assembled by PyMICE—Python library for mice behavioural data analysis [41]. For further analyses we used a critical part of this dataset consisting of the last 24 h of the place preference learning phase and next 24 h of reward-seeking or aversive discrimination learning phase [24].

## (d) Fear condition procedure

The mice were subjected to Pavlovian contextual fear conditioning in a fear conditioning chamber (MED Associates). The training was carried out according to a classic paradigm [25] and consisted of 3 min adaptation period and 5 footshocks lasting 1 s and having 0,6 mA intensity, which were applied with interstimulus intervals of 2 min. The animals were removed from the experimental cage to their home cages 2 min after the last footshock was applied. On the next day, the animals were tested in the same cage for 3 min. Fear to the context was assessed by measuring freezing behaviour for each individual animal. To avoid counting momentary inactivity as freezing, we scored an observation as freezing only if the mouse was immobile for at least 1 s. The freezing observations were transformed to a percentage of total observations in each of the 30 s intervals.

## (e) Morphometrics

Animals subjected to the award-seeking discrimination learning trial (figure 2a) were killed by cervical dislocation and dissected. Brain, heart, liver and kidneys were excised, blotted from excess fluids and weighed to an accuracy of 0.001 g.

## (f) LTP measurements

To gain insight into the neuronal mechanism underlying observed differences in learning we used LTP. We compared effects of repeated high-frequency stimulation of Schaeffer collaterals that make excitatory synapses onto pyramidal cells in the CA1 region of the hippocampus, the brain structure crucial for spatial memory formation [31,32]. For details see electronic supplementary material.

## (g) Statistical analyses

Data on BMR and masses of internal organs were analysed by means of ANCOVA with line type affiliation as a fixed factor, body mass as a covariate and the line type × body mass interaction. Initial BMR analyses also included the respirometric system and metabolic chamber coded as fixed factors. Their effects (as well as line type × body mass interaction) were never significant ($p > 0.05$), and therefore were dropped from final analyses.

In all analyses replicated lines were nested within line types as the random factor of the model (four replications in the RB and PMR line types, respectively, but 1 line for H-BMR and L-BMR line types, respectively, as they were not replicated; 10 lines in total).

Repeated-measures analysis of covariance (ANCOVA) was used to analyse the among-line type differences in total numbers of visits to all four corners summed over four continuous 12 h (dark followed by light) periods of observation ('period'), covering the last 24 h ('day') of place preference learning and first 24 h of reward-seeking discrimination learning.

In the reward-seeking, and aversive discrimination learning tasks we analysed the number of nosepokes to the bottles located in a corner assigned to a given animal during critical 48 h of trials. During the first two 12 h periods (located left of the vertical dashed line denoting the timeline of experiment, figure 2a–c) both bottles in the corner contained water. Subsequently, at the onset of the next two 12 h periods one bottle was filled with 10% sucrose (reward-seeking discrimination learning) or 0.005 M quinine solution (aversively motivated discrimination learning). In the ANCOVA model the numbers of correct responses (i.e. nosepokes to the bottle with sucrose) or incorrect responses (in case of quinine) were corrected for (1) the dark and the light experimental periods and the respective day (effects of both period and day coded as a fixed factors of a factorial design), and (2) the number of nosepokes to the bottle with tap water located in the same corner coded as a covariate. We used an analogously structured model to analyse the number of licks on the bottles containing tap or sweetened water.

To analyse the rate of changes in freezing response we used ANCOVA with the line type as a main factor, individual identification of animals and line (nested within line type) as random factors with time (subsequent 30 s intervals) as a covariate.

Data on LTP were analysed by means of repeated measures ANOVA with line type affiliation as a main factor. In this analysis, we compared the LTP slopes between the H-BMR and L-BMR line types along with one, randomly drawn RB line as the reference group. For further details of this and other analyses and their justification see electronic supplementary material.

All statistical analyses were carried out by means a mixed model extension of a general linear model ('Mixed' procedure of SAS/STAT 14.1 User's Guide) [42]. An SAS code of the 'Mixed' procedure is provided in the electronic supplementary material.

Our divergent selection on BMR is not replicated. Therefore, we cannot exclude that even the highly statistically significant differences between line types selected for BMR may stem from genetic drift, rather than from direct effects of the applied artificial selection. To evaluate the potential effect of genetic drift we therefore compared the magnitudes of the between-line type separation in key traits—internal organ mass, brain mass and the LTP with the ones expected under genetic drift. For detailed description of this analysis see electronic supplementary material.

Ethics. All procedures were approved by the Local Ethical Committee on Testing Animals.

Data accessibility. All row data are available from the Dryad Digital Repository (https://doi.org/10.5061/dryad.bk3j9kd78).

The data are provided in electronic supplementary material [43].

Authors' contributions. A.G.: conceptualization, investigation, visualization, writing—original draft, writing—review and editing; T.G.:

formal analysis, investigation, software; J.M.D.: formal analysis, investigation, software, visualization; J.J.-S.: formal analysis, investigation; E.K.: conceptualization, formal analysis, investigation, writing—original draft; M.K.: conceptualization, formal analysis, funding acquisition, investigation, writing—original draft, writing—review and editing.

All authors gave final approval for publication and agreed to be held accountable for the work performed therein.

Competing interests. We declare we have no competing interests.

Funding. This work was supported by National Research Center—NCN 2015/17/B/NZ8/02484 and 2017/27/B/NZ8/02242 grants.

Acknowledgements. We acknowledge valuable comments by Tomasz Burzykowski, Leszek Kaczmarek, Paweł Koteja, Jan Kozłowski, Krzysztof Mnich and an anonymous reviewer. Experimental assistance was kindly provided by A. Gębczyński, B. Lewończuk, M. Lewoc, S. Płonowski and J. Sadowska. Miłka Piszczek and Cleve Hicks greatly helped to edit the paper.

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
