## [Peer Review File · Proceedings of the Royal Society B: Biological Sciences]

Review History

RSPB-2021-1410.R0 (Original submission)

Review form: Reviewer 1

Recommendation

Reject – article is not of sufficient interest (we will consider a transfer to another journal)

Scientific importance: Is the manuscript an original and important contribution to its field?

Marginal

General interest: Is the paper of sufficient general interest?

Marginal

Quality of the paper: Is the overall quality of the paper suitable?

Marginal

Is the length of the paper justified?

Yes

Should the paper be seen by a specialist statistical reviewer?

No

Do you have any concerns about statistical analyses in this paper? If so, please specify them explicitly in your report.

Yes

It is a condition of publication that authors make their supporting data, code and materials available - either as supplementary material or hosted in an external repository. Please rate, if applicable, the supporting data on the following criteria.

Is it accessible?

No

Is it clear?

No

Is it adequate?

No

Do you have any ethical concerns with this paper?

No

Comments to the Author

RSPB-2021-1410

The authors use lines of mice selected for a higher metabolic rate to test the directionality of the 'Expensive Brain' (EB) hypothesis, which states that the energetic cost of evolving larger brains should be met with either a higher energy output or a reduced allocation of energy to other organs, and a reduction of the size of other organs, mostly the gut. The manuscript is well written, but I have concerns about how the results are presented, the significance of the results and the interpretation of them

1) My main concern is that there seems to be little logic to why mice with higher metabolic rates should have larger brains. I find the idea that a high metabolic rate could be a pre-requisite for a larger brain interesting, but is one that should be tested comparatively, not in a selection experiment. This makes the negative results presented not surprising. Then, the results are of morphological, behavioral and other differences between different metabolic lines, but none of them point to trade offs, which is presented as the main focus of the manuscript. While I think negative results are important, I don't think that in this case is unexpected. In other words, I think that the main conclusion, outlined in line 23, page 8. That "we revealed the likely directionality of the evolutionary relationships between energy expenditures, brain, gut and CA in a mammalian model of experimental evolution" is not supported by the results presented.

2) I have problems with the generalization made from the results. For example, the authors find differences in LTP between high and low metabolic rate lines and equate this to differences in brain plasticity. This is a long stretch, as the differences in the duration of the LTP may just reflect the differences in metabolism and not differences in brain plasticity. At most, the differences in LTP suggest differences in synaptic plasticity, but other evidence would be necessary to confirm this. Similarly, the authors equate differences in learning speed of two tasks with differences in cognitive abilities but alternative explanations, particularly related to food intake and motivation are possible.

3) Why only females were used? As the authors are likely aware, experiments that select for larger brains in fish have shown a strong sex effect. Is it possible that males show a change in brain size? Has this been tested?

4) Was the amount of food taken by each group quantified? It seems to me that the differences in organ size would could be explained by the amount of food taken by each group, as you would expect higher metabolic rate animals to take more food. Similarly, is not clear to me if this was tested in the reward seeking experiments. Is it possible that high metabolic rat lines just seek more food and therefore appear to learn faster?

5) Statistical test results should be added in the results, not only the tables. Also, the number of individuals is not reported in any experiment! Also effect size should be reported and result interpreted and discussed beyond a "significant" p value.

Minor issues:

1) Figure 1 would be better as a box plot with all data point showed. Colors of bar b and c are not distinct enough in black and white nor for color blind people (Authors should test this in a image processing software or use a tested color pallet)

Review form: Reviewer 2

Recommendation

Major revision is needed (please make suggestions in comments)

Scientific importance: Is the manuscript an original and important contribution to its field?

Excellent

General interest: Is the paper of sufficient general interest?

Excellent

Quality of the paper: Is the overall quality of the paper suitable?

Poor

Is the length of the paper justified?

Yes

Should the paper be seen by a specialist statistical reviewer?

No

Do you have any concerns about statistical analyses in this paper? If so, please specify them explicitly in your report.

Yes

It is a condition of publication that authors make their supporting data, code and materials available - either as supplementary material or hosted in an external repository. Please rate, if applicable, the supporting data on the following criteria.

Is it accessible?

Yes

Is it clear?

No

Is it adequate?

No

Do you have any ethical concerns with this paper?

No

Comments to the Author

The work concerns an important subject of general interest.

However, in our opinion the manuscript has several deficiencies that should – and can – be amended.

- 1) The text lacks several important information in methods and results, such as the number of animals used in the particular measurements (this omission disqualifies the manuscript by itself!), their age, several technical aspects of the measurements and of statistical analyses. Results of statistical analyses are not comprehensive, either (e.g., there is no information concerning random effects in the model, no estimates of slopes for quantitative predictors, no descriptive statistical information about the raw data – only adjusted means on graphs). If the overall length of the manuscript is an issue, an extended information could be perhaps provided in a supplement.
- 2) Even though the authors correctly underlie in Introduction the methodological strength of experimental evolution approach, they have not presented specific hypotheses to be tested (predictions concerning the expected results).
- 3) Limitations of the study resulting from the lack of replications of the H-BMR and L-BMR lines and the fact that two separate selection experiments are combined, and hence the RB lines are not strictly proper reference for the H- and L-BMR lines (so the hidden assumption that random variation among replicate lines in the second selection experiment applies also as source of random effects in the first one is doubtful) should be clearly acknowledged and addressed in Discussion. In the case of the analysis of neuronal plasticity the limitation is even stronger, because no information about the random effects could be applied.
- 4) Some aspects of the statistical data analysis need corrections.
- 5) Numerous references to literature are irrelevant, and this cannot be explained by just the simple confusion of the numbering (doubled number 1).
- 6) In our opinion the text is not clear in many places and requires many small amendments.
- 7) The raw dataset placed in repository does not contain all information needed to repeat the data analyses or to try alternative approaches.

Extensive specific comments are presented in the attached file.

Decision letter (RSPB-2021-1410.R0)

08-Sep-2021

Dear Dr Konarzewski:

I am writing to inform you that your manuscript RSPB-2021-1410 entitled "Brain size, gut size and cognitive abilities: the energy trade-offs tested in artificial selection experiment" has, in its current form, been rejected for publication in Proceedings B. This action has been taken on the advice of referees and the Associate Editor, who have recommended that substantial revisions are necessary. However, I, along with the reviewers and AE, feel that your manuscript could, with suitable revision, be an important addition to the literature. With this in mind we would be happy to consider a resubmission. Please note, however, that this is not a provisional acceptance; each of the reviewers' comments must be fully addressed and we will send the paper back out for review. Their comments can be found in full at the end of this email.

The resubmission will be treated as a new manuscript. However, we will approach the same reviewers if they are available and it is deemed appropriate to do so by the Editor. Please note that resubmissions must be submitted within six months of the date of this email. In exceptional

circumstances, extensions may be possible if agreed with the Editorial Office. Manuscripts submitted after this date will be automatically rejected.

Sincerely,
 Dr Sarah Brosnan
 Editor, Proceedings B
 mailto: proceedingsb@royalsociety.org

Associate Editor
 Board Member: 1
 Comments to Author:

This study presents an experimental approach on a mammalian animal model in which the effect of artificial selection for energy expenditure on cognitive abilities and neuronal plasticity are tested. The findings are intriguing: with higher energetic expenditure comes higher cognitive abilities and higher synaptic plasticity, but not higher brain size.

I agree with the authors that the results suggest that an effect of energy expenditure may be present. However, in its present form, the study leaves open too many uncertainties. Both reviewers highlight potential confounding aspects that are likely to influence the results. For example, reviewer 1 highlights sex and food intake, and reviewer 2 additionally highlights the age of the animals, the lack of replications of the H-BMR and L-BMR lines, and the fact that two separate selection experiments are combined.

Both reviewers also note that information on several important aspects of the research design are missing (e.g., the number of individuals used for particular measurements and crucial statistical details of several analysis).

In addition to these methodological and statistical concerns, there are also some concerns with regards to the justification of the study and the interpretation of the results. A more precise account of why the authors expect mice with higher metabolic rates to have larger brains would be needed to strengthen the justification of the research question. I further agree with reviewer 1 that the leap from differences in LTP between high and low metabolic rate lines to brain differences in brain plasticity requires more justification.

Although I recognize that there appear to be several opportunities to improve the research design to meet these concerns, the manuscript is not currently acceptable to be considered for publication in Proc B. If such improvements to the research design can be implemented and such changes improve the accuracy of the results, I would like to encourage the authors to resubmit.

Reviewer(s)' Comments to Author:

Referee: 1

Comments to the Author(s)

RSPB-2021-1410

The authors use lines of mice selected for a higher metabolic rate to test the directionality of the 'Expensive Brain' (EB) hypothesis, which states that the energetic cost of evolving larger brains should be met with either a higher energy output or a reduced allocation of energy to other organs, and a reduction of the size of other organs, mostly the gut. The manuscript is well written, but I have concerns about how the results are presented, the significance of the results and the interpretation of them

1) My main concern is that there seems to be little logic to why mice with higher metabolic rates should have larger brains. I find the idea that a high metabolic rate could be a pre-requisite for a larger brain interesting, but is one that should be tested comparatively, not in a selection experiment. This makes the negative results presented not surprising. Then, the results are of morphological, behavioral and other differences between different metabolic lines, but none of them point to trade offs, which is presented as the main focus of the manuscript. While I think negative results are important, I don't think that in this case is unexpected. In other words, I think that the main conclusion, outlined in line 23, page 8. That "we revealed the likely directionality of the evolutionary relationships between energy expenditures, brain, gut and CA in a mammalian model of experimental evolution" is not supported by the results presented.

2) I have problems with the generalization made from the results. For example, the authors find differences in LTP between high and low metabolic rate lines and equate this to differences in brain plasticity. This is a long stretch, as the differences in the duration of the LTP may just reflect the differences in metabolism and not differences in brain plasticity. At most, the differences in LTP suggest differences in synaptic plasticity, but other evidence would be necessary to confirm this. Similarly, the authors equate differences in learning speed of two tasks with differences in cognitive abilities but alternative explanations, particularly related to food intake and motivation are possible.

3) Why only females were used? As the authors are likely aware, experiments that select for larger brains in fish have shown a strong sex effect. Is it possible that males show a change in brain size? Has this been tested?

4) Was the amount of food taken by each group quantified? It seems to me that the differences in organ size could be explained by the amount of food taken by each group, as you would expect higher metabolic rate animals to take more food. Similarly, it is not clear to me if this was tested in the reward seeking experiments. Is it possible that high metabolic rat lines just seek more food and therefore appear to learn faster?

5) Statistical test results should be added in the results, not only the tables. Also, the number of individuals is not reported in any experiment! Also effect size should be reported and result interpreted and discussed beyond a "significant" p value.

Minor issues:

1) Figure 1 would be better as a box plot with all data points shown. Colors of bar b and c are not distinct enough in black and white nor for color blind people (Authors should test this in an image processing software or use a tested color palette)

Referee: 2

Comments to the Author(s)

The work concerns an important subject of general interest.

However, in our opinion the manuscript has several deficiencies that should – and can – be amended.

- 1) The text lacks several important information in methods and results, such as the number of animals used in the particular measurements (this omission disqualifies the manuscript by itself!), their age, several technical aspects of the measurements and of statistical analyses. Results of statistical analyses are not comprehensive, either (e.g., there is no information concerning random effects in the model, no estimates of slopes for quantitative predictors, no descriptive statistical information about the raw data – only adjusted means on graphs). If the overall length of the manuscript is an issue, an extended information could be perhaps provided in a supplement.
- 2) Even though the authors correctly underlie in Introduction the methodological strength of experimental evolution approach, they have not presented specific hypotheses to be tested (predictions concerning the expected results).
- 3) Limitations of the study resulting from the lack of replications of the H-BMR and L-BMR lines and the fact that two separate selection experiments are combined, and hence the RB lines are not strictly proper reference for the H- and L-BMR lines (so the hidden assumption that random variation among replicate lines in the second selection experiment applies also as source of random effects in the first one is doubtful) should be clearly acknowledged and addressed in Discussion. In the case of the analysis of neuronal plasticity the limitation is even stronger, because no information about the random effects could be applied.
- 4) Some aspects of the statistical data analysis need corrections.
- 5) Numerous references to literature are irrelevant, and this cannot be explained by just the simple confusion of the numbering (doubled number 1).
- 6) In our opinion the text is not clear in many places and requires many small amendments.
- 7) The raw dataset placed in repository does not contain all information needed to repeat the data analyses or to try alternative approaches.

Extensive specific comments are presented in the attached file.

Author's Response to Decision Letter for (RSPB-2021-1410.R0)

See Appendix A.

RSPB-2021-2747.R0

Review form: Reviewer 2

Recommendation

Major revision is needed (please make suggestions in comments)

Scientific importance: Is the manuscript an original and important contribution to its field?

Excellent

General interest: Is the paper of sufficient general interest?

Excellent

Quality of the paper: Is the overall quality of the paper suitable?

Marginal

Is the length of the paper justified?

Yes

Should the paper be seen by a specialist statistical reviewer?

No

Do you have any concerns about statistical analyses in this paper? If so, please specify them explicitly in your report.

Yes

It is a condition of publication that authors make their supporting data, code and materials available - either as supplementary material or hosted in an external repository. Please rate, if applicable, the supporting data on the following criteria.

Is it accessible?

Yes

Is it clear?

Yes

Is it adequate?

Yes

Do you have any ethical concerns with this paper?

No

Comments to the Author

General comment

The authors responded adequately to majority of our comments, and the manuscript is now much improved. However, some of the deficiencies, e.g. inconsistent terminology, were corrected only in the places we indicated directed, rather than in the entire manuscript. In a few cases, in which the authors chose to not follow our advice, we do not agree with their arguments, and we now explain the points more explicitly. Most importantly, a few important aspects of the statistical analyses remained not corrected. Unlike in the cases of presentation style or the results interpretation, in which we respect the authors right to stay with their opinion, in these few cases it is a matter of presenting results that are legitimate or not. Importantly, however, even though our list of comments is again extensive, we are confident that the authors can resolve the issues and provide a really valuable contribution, and the next revision of the work could be rated even as excellent. Also importantly, this opinion will not change even if the revised analyses would lead to somewhat different conclusions.

Detailed comments are in the attached file. (See Appendix B)

Decision letter (RSPB-2021-2747.R0)

18-Feb-2022

Dear Dr Konarzewski:

Your revised manuscript has now been peer reviewed and the review has been assessed by an Associate Editor. The reviewer, AE, and I particularly appreciate the care that you took on the previous round of revision and agree that your manuscript has the potential to be a very strong contribution. As you will see, however, the reviewer has highlighted a few instances in which

there are still some misunderstandings about what the reviewer intended, so we invite you to further revise your manuscript to clarify these points. The reviewers' comments (not including confidential comments to the Editor) and the comments from the Associate Editor are included at the end of this email for your reference.

Research ethics:

Use of animals and field studies:

It is a condition of publication that you make available the data and research materials supporting the results in the article (<https://royalsociety.org/journals/authors/author-guidelines/#data>). Datasets should be deposited in an appropriate publicly available repository and details of the associated accession number, link or DOI to the datasets must be included in the Data Accessibility section of the article (<https://royalsociety.org/journals/ethics-policies/data-sharing-mining/>). Reference(s) to datasets should also be included in the reference list of the article with DOIs (where available).

Please submit a copy of your revised paper within three weeks. If we do not hear from you within this time your manuscript will be rejected. If you are unable to meet this deadline please let us know as soon as possible, as we may be able to grant a short extension.

Best wishes,
Dr Sarah Brosnan
Editor, Proceedings B
mailto: proceedingsb@royalsociety.org

Associate Editor
Comments to Author:

I would like to thank the authors for incorporating reviewer comments in such a thorough manner and for resubmitting the manuscript. We sent the resubmission out for reviewer to one reviewer only. We are now all in agreement that the contribution of this paper is excellent and we would like to see this paper move forward in the publication process. Importantly though, there appears to have been some level of miscommunication between the reviewer comments and the authors' reply to these comments in the previous submission. The reviewer highlights several such miscommunications, and phrases his/her comments more explicitly. I agree with the reviewer that these comments are important and will need to be addressed.

Reviewer(s)' Comments to Author:

Referee: 2

Comments to the Author(s).

General comment

The authors responded adequately to majority of our comments, and the manuscript is now much improved. However, some of the deficiencies, e.g. inconsistent terminology, were corrected only in the places we indicated directed, rather than in the entire manuscript. In a few cases, in which the authors chose to not follow our advice, we do not agree with their arguments, and we now explain the points more explicitly. Most importantly, a few important aspects of the statistical analyses remained not corrected. Unlike in the cases of presentation style or the results interpretation, in which we respect the authors right to stay with their opinion, in these few cases

it is a matter of presenting results that are legitimate or not. Importantly, however, even though our list of comments is again extensive, we are confident that the authors can resolve the issues and provide a really valuable contribution, and the next revision of the work could be rated even as excellent. Also importantly, this opinion will not change even if the revised analyses would lead to somewhat different conclusions.

Detailed comments are in the attached file

Author's Response to Decision Letter for (RSPB-2021-2747.R0)

See Appendix C.

Decision letter (RSPB-2021-2747.R1)

21-Mar-2022

Dear Dr Konarzewski

I am pleased to inform you that your manuscript entitled "Brain size, gut size and cognitive abilities: the energy trade-offs tested in artificial selection experiment" has been accepted for publication in Proceedings B.

Data Accessibility section

Open Access

Paper charges

Sincerely,

Dr Sarah Brosnan

Appendix A

Response to Referees

Brain size, gut size and cognitive abilities: the energy trade-offs tested in artificial selection experiment

by

Anna Goncerzewicz, Tomasz Górkiewicz, Jakub M. Dzik, Joanna Jędrzejewska-Szmek,
Ewelina Knapska and Marek Konarzewski

We thank the reviewers for their in-depth comments. Accordingly, our submission has been revised in the following key areas:

1. We have now provided a concise description of specific predictions.
2. We have addressed head- on the issue of the lack of replication of our divergent selection for BMR. To evaluate the potential effect of genetic drift we compared the magnitude of the between- line type separation in key traits— internal organ mass, brain mass and the LTP — with the one expected under genetic drift. We provided detailed description of this analysis in Supplementary Materials.
3. In the main text and Supplementary Materials we have now provided detailed description of data handling and statistical analyses. Furthermore, we have re-analysed our major statistical models using a factorial design approach, as suggested by the Referees #2. Qualitatively, it did not change our conclusions.

To assist the Referees, we provide a clean amended copy of our submission along with a copy with parts of the text (marked in green) affected by the revision. In our responses to the Referees' comment given below we refer in an itemised fashion to specific lines of the text.

Referee: 1

1) My main concern is that there seems to be little logic to why mice with higher metabolic rates should have larger brains. I find the idea that a high metabolic rate could be a prerequisite for a larger brain interesting, but is one that should be tested comparatively, not in a selection experiment. This makes the negative results presented not surprising. Then, the results are of morphological, behavioral and other differences between different metabolic lines, but none of them point to trade offs, which is presented as the main focus of the manuscript. While I think negative results are important, I don't think that in this case is unexpected. In other words, I think that the main conclusion, outline in line 23, page 8. That "we revealed the likely directionality of the evolutionary relationships between energy expenditures, brain, gut and CA in a mammalian model of experimental evolution" is not supported by the results presented.

Our response: We realise that Referee's comments, if justified, provide a good reason to reject our paper. We therefore treat them with utmost attention. Referee alludes to separate, albeit related issues. Below we therefore quote specific comments and address them one by one.

'My main concern is that there seems to be little logic to why mice with higher metabolic rates should have larger brains.'

Our response: We respectfully disagree. First, high basal metabolic rate is functionally linked to brain size because of an exceptionally high metabolic rate of brain tissue- an order of magnitude higher than in, e.g. resting muscles (Elia 1992). The metabolic rate of nervous tissue is high even at rest because of the need to maintain the membrane electrochemical potential (Kuzawa et al., 2014) and other processes that are not fully understood but are independent of external stimuli (Raichle, 2006). Thus, larger brains inevitably incur high metabolic costs reflected in BMR. This is best exemplified by exceptionally large brains of adult humans, whose 20-25% of BMR is accounted for by brain metabolism, even though brain tissue accounts for only 2% of adult body mass (Javed et al. 2010, Pontzer et al. 2021). Likewise, this positive brain size- BMR relationship holds in rodents (Sobrero et al. (2011)., and other phyla (e.g., fish- Sukhum et al. 2016). This allowed us to found our paper on a straightforward prediction of a positive, functional association between BMR and brain size, which is well grounded in literature (see also Armstrong 1983).

‘I find the idea that a high metabolic rate could be a pre-requisite for a larger brain interesting, but is one that should be tested comparatively, not in a selection experiment. This makes the negative results presented not surprising.’

Our response: We respectfully disagree. We surmise that the Referee alludes to comparative analyses on the between-species level, and considers studies on intraspecific level not sufficiently informative. It must be borne in mind, however, that it is intra-specific variation that is a substrate of natural selection, and therefore, inter-specific studies can only partially inform the inference on adaptation and cannot unambiguously identify factors influencing its variation. For this reason, patterns derived from inter-specific analyses cannot be directly extrapolated to the intra-specific level. For example, Herculano- Houzel et al. demonstrated that cellular brain allometry within outbred mouse strains is not simply an extension of cellular brain composition at interspecific level (Herculano Houzel et al. 2015). Furthermore, brain size and composition is heritable (e.g., Atchley et al. 1984) and responsive to artificial selection (also in mice, e.g., Perepelkina et al. 2013). This points to experimental evolution as a viable option for testing the directionality of the evolutionary relationships involving brain size.

‘Then, the results are of morphological, behavioral and other differences between different metabolic lines, but none of them point to trade- offs, which is presented as the main focus of the manuscript. While I think negative results are important, I don't think that in this case is unexpected.’

Our response: We would like to point out that we framed identification of the trade- offs within much broader context, as we aimed at testing whether ‘enlarged brain and the resulting increased cognitive abilities (CA) were met either by increased energy turnover or reduced allocation to other expensive organs, such as the gut.’ (lines 3-5 of the Abstract). Thus a possible trade- off between brain size and the gut was only one of the tested alternatives, not the ‘main focus of the paper’. Indeed, our results do not support the existence of such a trade-off, but taking into consideration our arguments on brain size evolvability given above, we cannot agree that this result is ‘unexpected’.

Referee’s comments partly correspond with the comments of other Referees. They have prompted us to rewrite several parts of the Abstract (p. 2, lines 6-10) Introduction (p. 3, lines 11-15) and Discussion (p. 7, line 34, p. 8, lines 1-3). Furthermore, we have carried out

additional analyses to demonstrate that the ‘negative result’ of the lack of the brain size- gut trade- off stems from a weak response of brain size to selection on BMR, in contrast to other internal organs (section ‘Evaluation of the effect of genetic drift in divergent selection for BMR’ in Supplementary Materials).

References to the above responses:

Armstrong E. 1983. Relative brain size and metabolism in mammals. *Science* 220, 1302–1304.

(doi:10.1126/science.6407108).

Atchley William R., Bruce Riska, Luci A. P. Kohn, A. Alison Plummer and J. J. Rutledge. 1984. A Quantitative Genetic Analysis of Brain and Body Size Associations, Their Origin and Ontogeny: Data From Mice *Evolution* 38: 1165-1179.

Elia M. 1992. Organ and tissue contribution to metabolic rate. In: Kinney JM, Tucker HN, eds. *Energy metabolism: tissue determinants and cellular corollaries*. New York, NY: Raven Press, Ltd, 1992:61–79.

Herculano-Houzel Suzana, Débora J. Messeder, Karina Fonseca-Azevedo and Nilma A. Pantoja. 2015. When larger brains do not have more neurons: increased numbers of cells are compensated by decreased average cell size across mouse individuals. *Front. Neuroanat.*, <https://doi.org/10.3389/fnana.2015.00064>

KUZAWA, C.W., CHUGANI, H. T., GROSSMAN, L. I., LIPOVICH, L., MUZIK, O., HOF, P.R., WILDMAN, D. E., SHERWOOD, C. C., LEONARD, W. R. & LANGE, N. (2014). Metabolic costs and evolutionary implications of human brain development. *Proceeding of the National Academy of Sciences USA* 111, 13010–13015.

Perepelkina, O.V., Golibrodov, V.A., Lilp, I.G., Poletaeva, I.I. 2013. Mice selected for large and small brain weight: The preservation of trait differences after the selection was discontinued. *Advances in Bioscience and Biotechnology*. 4:1-8.

RAICHLE, M. E. (2006). The brain’s dark energy. *Science* 314, 1249–1250.

Sobrero et al. (2011). Expensive brains: "brainy" rodents have higher metabolic rate. *Frontiers in Evolutionary Neuroscience* 3,2:1-12.

Sukhum KV, Freiler MK, Wang R, Carlson BA. 2016. The costs of a big brain: extreme encephalization results in higher energetic demand and reduced hypoxia tolerance in weakly electric African fishes.

Proc. R. Soc. B 283: 20162157. <http://dx.doi.org/10.1098/rspb.2016.2157>

2) *I have problems with the generalization made from the results. For example, the authors find differences in LTP between high and low metabolic rate lines and equate this to differences in brain plasticity. This is a long stretch, as the differences in the duration of the LTP may just reflect the differences in metabolism and not differences in brain plasticity. At most, the differences in LPT suggest differences in synaptic plasticity, but other evidence would be necessary to confirm this. Similarly, the authors equate differences in learning speed of two tasks with differences in cognitive abilities but alternative explanations, particularly related to food intake and motivation are possible.*

Our response: Long-term potentiation (LTP) of excitatory synaptic transmission has long been recognized as a cellular correlate of learning and memory (Nicoll 2017). Both, molecular and cellular components of the LTP and its relationship to learning and memory are well described and understood (Nicoll 2017). Early LTP, that we studied here, requires phosphorylation of AMPA receptors. In particular, it has been shown that interfering with AMPAR surface diffusion impairs both synaptic potentiation of Schaffer collaterals and commissural inputs to the CA1 area of the mouse hippocampus and inhibits contextual fear conditioning (Penn et al., 2017). Phosphorylation of AMPA receptors is ATP dependent (Banke et al. 2000), and therefore should be positively associated with the rate of aerobic metabolism. Therefore, we disagree with the Reviewer that ‘the differences in the duration of the LTP may just reflect the differences in metabolism and not differences in brain plasticity’. We have now clarified this issue on p. 7, in lines 25-30 of Discussion.

We relegate our response to the query on food intake and motivation to the comment below, directly related to this problem.

References to the above response:

Nicoll, R.A. 2017. A Brief History of Long-Term Potentiation. *Neuron*:
DOI:<https://doi.org/10.1016/j.neuron.2016.12.015>

Penn AC, Zhang CL, Georges F, Royer L, Breillat C, Hosy E, Petersen JD, Humeau Y, Choquet D. Hippocampal LTP and contextual learning require surface diffusion of AMPA receptors. *Nature*. 2017 Sep 21;549(7672):384-388. doi: 10.1038/nature23658.

Banke, T. G., D. Bowie, H.-K. Lee, R. L. Huganir, A. Schousboe, and S. F. Traynelis 2000. Control of GluR1 AMPA Receptor Function by cAMP-Dependent Protein KinaseJ. *Neurosci.*, 20(1):89–102.

3) *Why only females were used? As the authors are likely aware, experiments that select for larger brains in fish have shown a strong sex effect. Is possible that males show a change in brain size? Has this been tested?*

Our response: We used females to avoid agonistic interactions often manifested by male mice (Lidster et al. 2019). Please note that our behavioural experiments in IntelliCage system involved maintenance of groups of individual mice. Therefore, a significant level of between-individual aggression, which is much higher in males, would impede on the results of our study. We have now clarified this in lines 15-20 of the Supplementary Materials.

Lidster K, Owen K, Browne WJ, Prescott MJ. Cage aggression in group-housed laboratory male mice: an international data crowdsourcing project. *Sci Rep*. 2019 Oct 23;9(1):15211. doi: 10.1038/s41598-019-51674-z.

4) *Was the amount of food taken by each group quantified? It seems to me that the differences in organ size would be explained by the amount of food taken by each group, as you would expect higher metabolic rate animals to take more food. Similarly, is not clear to me if this was tested in the reward seeking experiments. Is it possible that high metabolic rate lines just seek more food and therefore appear to learn faster?*

Our response: No, we did not quantify food intake. However, in several earlier studies we demonstrated that mice of the H-BMR line are characterized by higher food consumption and internal organs larger than in other studied line types (Sadowska et al. 2019 and references therein). We have also demonstrated that this high food consumption and relatively larger

internal organs are genuine outcomes of the artificial selection for high BMR, rather than random effects of genetic drift (Sadowska et al. 2019). So indeed, we agree with the Referee that the differences in organs size depicted in Fig. 1 most likely reflect the differences in food consumption. Importantly, however, this is not merely a trivial phenotypic effect, but rather manifestation of genetic correlations arising due to the applied selection on BMR (see Figure S1 in Supplementary Materials).

In our behavioural experiments we used sweetened water, not food, as a reward. Food was available ad libitum all the time, so for mice this was not a learning cue. Therefore, we see no reason to expect that food intake would somehow affect our results. Also, please note that we did not observe any differences between the line types in the amount of sweetened water consumed (gauged as the number of licks, Table 2). Thus, its nutritional effect on the outcomes of behavioural tests can be ruled out.

Since the number of licks, which are a proxy for the amount of consumed liquid, and general activity were comparable between the line types, we argue that the difference is limited to operant responses (nosepokes) that lead to obtaining sweetened water or bitter quinine solution. The H-BMR animals respond more robustly than the mice of other line types, to changing environmental cues by increasing the number of nosepokes to the reward and decreasing it to the quinine solution, which is an adaptive response.

Julita Sadowska, Andrzej K. Gębczyński, Małgorzata Lewoc and Marek Konarzewski. Not that hot after all: no limits to heat dissipation in lactating mice selected for high or low BMR. (2019) *Journal of Experimental Biology* 222, doi:10.1242/jeb.204669

5) Statistical test results should be added in the results, not only the tables. Also, the number of individuals is not reported in any experiment! Also effect size should be reported and result interpreted and discussed beyond a "significant" p value.

Our response: The targeted journal has a very concise form and limited text length. For this reason we have reduced the information on statistical methods and results to minimum. We agree that by doing so we failed to provide several important information. To remedy this problem we have now provided a full account of statistical methods in the Supplementary Materials. The numbers of animals are presented in Table S1.

As for the size effect, please note that most important parts of our reasoning are based on very conservative numbers of degrees of freedom, with the df for the between line type comparisons equal to 3 (for the F numerator) and 6 (for denominator), respectively. Furthermore, our experimental design is based on replications with dfs representing sublines, not individual animals. Therefore, recent concerns (e.g., Ranganathan et al. 2015) related to the robustness of inference based on the p value do not directly apply to here, with the exception of non-significance of the between-line difference in brain mass, rightly pointed out by the referee in his/her comment (1). This potential issue is particularly important because our divergent selection on BMR is not replicated, so the lack of selection response in secondary trait (brain mass) judged by the p value does not provide strong inference. To tackle this problem we have now compared the differences between the H-BMR and L-BMR line types in the LTP, brain mass and other studied organs with the ones expected to arise randomly, that is under genetic drift (see Henderson 1997, Konarzewski et al. 2005 now cited in the Supplementary Materials). We have demonstrated that a weak separation of the H-BMR and L-BMR line types with respect to brain mass does not allow for its firm recognition as a correlated response to selection on BMR, so the effect of genetic drift cannot be ruled out. In contrast, the between line type differences in the LTP and internal organ masses are genuine effects of the applied selection regimen and indicate their positive genetic

correlations with BMR. For the sake of brevity we have now concisely presented these results in the main body of paper (p. 4, lines 15-22), relegating their full description to the Supplementary Materials.

Ranganathan P, Pramesh CS, Buyse M. 2015. Common pitfalls in statistical analysis: "P" values, statistical significance and confidence intervals. *Perspect. Clin. Res* 6:116-7.

Minor issues:

Figure 1 would be better as a box plot with all data point showed. Colors of bar b and c are not distinct enough in black and white nor for color blind people (Authors should test this in a image processing software or use a tested color pallet).

Our response: We elected to leave Figure 1 as a bar chart. However, now have now added Figure S2 in the Supplementary Materials, which is a box plot of data used to compose Figure 1. We also changed the colors in Figure 1.

Referee(s): 2

1) *The text lacks several important information in methods and results, such as the number of animals used in the particular measurements (this omission disqualifies the manuscript by itself!), their age, several technical aspects of the measurements and of statistical analyses. Results of statistical analyses are not comprehensive, either (e.g., there is no information concerning random effects in the model, no estimates of slopes for quantitative predictors, no descriptive statistical information about the raw data – only adjusted means on graphs). If the overall length of the manuscript is an issue, an extended information could be perhaps provided in a supplement.*

Our response: Yes, we admit that because of a concise style of the targeted journal we have not detailed several intricacies of statistics and methods. We have now extended methods description in the main text and provided further information in Supplementary Materials, as detailed below, in our responses to subsequent queries.

2) *Even though the authors correctly underlie in Introduction the methodological strength of experimental evolution approach, they have not presented specific hypotheses to be tested (predictions concerning the expected results).*

Our response: On p. 3 lines 30-31 we have now provided a concise description of specific aims and predictions. See also our response to the first of the specific comments of the Referee(s).

3) *Limitations of the study resulting from the lack of replications of the H-BMR and L-BMR lines and the fact that two separate selection experiments are combined, and hence the RB lines are not strictly proper reference for the H- and L-BMR lines (so the hidden assumption that random variation among replicate lines in the second selection experiment applies also as source of random effects in the first one is doubtful) should be clearly acknowledged and addressed in Discussion. In the case of the analysis of neuronal plasticity the limitation is even stronger, because no information about the random effects could be applied.*

Our response: We have explicitly indicated that our selection on BMR is not replicated (p. 8, line 12). However, we agree that this weakness has not been fully acknowledged and addressed. We have now carried out an additional analyses of the L-BMR vs. H-BMR differences in key traits—brain mass, internal organ masses and LTP evaluating their

magnitude against the differences arising randomly, due to genetic drift. For further details please see our response to similar comment by Referee #1.

4) *Some aspects of the statistical data analysis need corrections.*

5) *Numerous references to literature are irrelevant, and this cannot be explained by just the simple confusion of the numbering (doubled number 1).*

6) *In our opinion the text is not clear in many places and requires many small amendments.*

7) *The raw dataset placed in repository does not contain all information needed to repeat the data analyses or to try alternative approaches.*

Our response: We have addressed remarks 4) - 7) in our responses to specific comments below.

Specific comments

Page 2, lines 6-10

The important information about both the general objective of the work and the animal model is presented in one, very long and complex sentence, which is difficult to follow and confusing. What do you mean by “the directionality of the evolutionary relationships between energy expenditures, brain, gut and CA”? In fact, you tested for the presence and direction of correlated responses to selection, which is equivalent (roughly) to testing the presence and direction of genetic correlation between the traits. Note, that this phrase could be substituted by just telling that you „tested the EB hypothesis,” because the same specific traits are mentioned in the sentence presenting the hypothesis. The phrase „using an experimental evolution model in which we subjected line types of laboratory mice to artificial selection on basal (BMR) or maximum (VO₂max) aerobic metabolism...” is also unclear and not very informative. It suggests that you started with some „line types” (which implies that they somehow differed already on the onset of the experimental evolution project), and then applied selection for BMR and VO₂max in each of the line types. The sentence does not explain what was the direction of the selection. As a consequence further text of the Abstract, reporting results, is unclear, too. We suggest to split the long sentence, and provide more explicit information, e.g.:

We tested the EB hypothesis by analyzing correlated responses to selection in an experimental evolution model system, which comprises four line types of laboratory mice: selected for high or low basal (BMR) and high maximum (VO₂max) metabolic rates, and unselected control. The traits are implicated in evolution of homeothermy, having been pre-requisites for the encephalisation and exceptional CA of mammals, including humans.

Note, that the suggested text has a smaller number of characters than the original version, but is more informative and easier to read. This is not the only place in which the clearness of the text can be improved in a similar way. We suggest to revise the entire text considering this hint.

Our response: Thank you. We have modified the Abstract accordingly and attempted to apply this this excellent hint to the remaining text.

Page 2, lines 10-14:

Further part of Abstract refers results only for High BMR lines, Moreover, the text always says they had some values “higher,” but not “higher than what.” Higher than all other line types, or the low BMR, or the unselected? Results concerning other comparisons (even if negative), should be also mentioned.

Our response: We have now specified the directionality of the between-line type differences. We are unable to report the results of other comparisons due to a strict word limit imposed on the Abstract length (200 words).

Page 2, line 22 and further:

You have doubled the first number on the list of references, and therefore all further reference numbers are incorrect.

Our response: Corrected.

Page 2, Lines 28-30

The experimental results of the cited paper [8, but should be 9] mentioned not only gut size decrease, but also decreased reproductive performance, as results of the selection for the increased brain size. This finding appears in one more reference [13, Isler and van Schaik, 2009] and Weisbecker and Goswami 2010 present similar findings. We suggest to mention this aspect, too.

Our response: We have now briefly touched upon this important issue on p. 3, lines 11-15. (NB., please note that throughout our responses we refer to page numbering of the manuscript, not the numbering created in the process of generation of PDF file).

Page 3, line 8

Missing comma after “However.” Also, the text is a continuation of a discussion presented in the previous paragraph, and hence dividing this part of the text into two separate paragraphs does not seem necessary.

Our response: Corrected.

Page 3, line 14

The citations 18-20 (17-19 in ref list) actually did not show a positive link between evolution of cognitive abilities or brain size and high aerobic exercise performance. Koteja 2004 is a minireview/ concept study, in which such a relation was perhaps mentioned, but not documented. Chrzęścik et al. 2014 actually showed NO increase of learning capacity in voles selected for high aerobic performance, although it refers to a conference presentation reporting increased brain size in the selected voles. The last of the three, Książek et al. 2009, is quite irrelevant in the context of the sentence – as the authors should be well aware of. Thus, by all appearances, there was some mix-up with the literature and all the text should be carefully checked for the relevance of the citations.

Our response: Yes, we admit that references cited in this context were not appropriate. We now cite Raichlen DA, Gordon AD. 2011 (directly demonstrating a positive correlation between brain size and maximum metabolic rate in mammals, MMR) and Hayes et al. 2018 showing a positive correlation between MMR and geographic range size in mammals.

Page 3, lines 21-26

The closing sentence of the first of the two paragraphs and the opening sentence of the next one imply that no relevant research based on experimental evolution on mammalian models has been previously undertaken. However, Kolb et al 2013 analyzed brain size in mice selected for high-wheel running, and Wikgren et al. 2012 analyzed cognitive abilities in rats

divergently selected for endurance running – a trait closely related to the aerobic capacity indicated as important trait here. Chrzęścik et al. 2014 analyzed learning abilities in voles from a multidirectional selection experiment, and refer to a conference abstract with results showing increased brain size in lines selected for high aerobic metabolism. The first two of these papers were not mentioned at all, and Chrzęścik et al. is referred to, but in an irrelevant way, and without indicating that this was a work based on selection experiment. It will not decrease the value of the work if you recognize that other researchers have also seen the potential of experimental evolution approach on mammalian models in the context of the relation between the brain size or cognitive abilities evolution and metabolic traits, even if none of these considered simultaneously all the aspects present in your work (so you can still say that it is “the first time”).

Our response: We totally agree, and we also appreciate the significance and merit of the above mentioned papers. However, none of these studies directly aimed at testing the EB hypothesis. Because of a concise style of the targeted journal, we cannot appropriately refer to all of them. We have shortened the sentence in question, to directly allude to the EB hypothesis.

Page 3, line 26-29

As in Abstract, the sentence is not clear and the phrase “line types” sounds awkward. We understand the necessity to differentiate between the replicate lines and line types, but it can be phrased in a more comprehensible manner. The “replicate lines” within line types can be briefly mentioned in this paragraph, which should help the reader realize why are you not talking simply about “lines.” In addition, the phrase “selection on divergent rates of basal (BMR) or maximum aerobic metabolism VO2max” is misleading, because it suggests that VO2max was also divergently selected.

Our response: Many years ago we borrowed the terms ‘line’ and ‘line type’ from papers published by Ted Garland’s group, and we use this notation in subsequent publications ever since. For this reason we would like to stick with it. Unfortunately, because of a strict length limitation of the Abstract section (no more than 200 words) we elected to relegate the explanation of this notation to Materials and Methods section. We amended the text to clarify the directionality of artificial selection.

Page 3 lines 29-33

In this sentence two completely different issues are mixed: the difference between correlative comparative analyses vs. the experimental evolution approach (OK), and the possibility to test association between metabolic traits and learning capabilities – which is not dependent on applying the experimental evolution approach. Better separate these logically unrelated aspects.

Our response: We agree. We deleted the first part of the sentence to clarify argumentation.

Page 4 lines 1-8

You introduce the issue of the association between brain size and the cognitive capability only at the very end of Introduction, after declaring the specific objectives in the previous paragraph. Perhaps it would be better to raise the issue earlier. Particularly, it is relevant in the paragraph in which you provide the rationale for the first scenario of EB – that the larger brain allows more efficient foraging.

Our response: We hope that the changes introduced on p. 3, in lines 11-15 addressed this comment.

Page 4 and further

No information about the number of animals/observations is provided, either in Results or Method sections.

Our response: In Table S1 of Supplementary Materials we have now provided full information on the numbers.

Page 4 line 11

We still do not have a clear information about the line types, not even how many line types are “used” (unselected control has not been mentioned yet). We also do not know what is compared (means, medians, ranges?). The information is provided in a complicated way. What do you mean by “with respect to body mass”? Do you simply mean: “Mean body mass did not differ between four line types”?

Our response: Detailed information on lines, line types etc. is provided in Materials and Methods section located below the main text (in compliance with the journal’s ‘Author Guidelines’) and Supplementary Materials, where we moved technicalities, to meet the length restrictions. Yes, we mean ““Mean body mass did not differ among mice from four line types” (corrected).

Page 4, lines 11-12

“Yet, high BMR mice ... were characterized by higher BMR” Why “yet”? In what way the lack of differences in body mass makes it particularly impressive that BMR was higher in lines of mice selected for high BMR? The sentence is misleading also because it suggests that differences among other lines are less conspicuous. However, the difference between the L-BMR and RB lines is in fact even larger than that between H-BMR and the RB lines.

Our response: Yes thank you. We deleted ‘Yet’.

Page 4, Line 12

Please inform already here that the reported (and selected for) BMR is a mass-corrected value. We know it is explained in Methods, but adding the two words does not cost much, and is helpful in the scheme in which Methods are pushed to the end of text.

Our response: Yes, we have added the phrase ‘mass-corrected’ in several places throughout the text.

Page 4, lines 11-15

Why does the entire paragraph concentrate only on one line type? The letters on Figure 1 indicate that there are more differences between line types, which are not mentioned in the text. The results and discussion in general concentrate only on H-BMR line, hardly ever mentioning even as much as that the effects of selection on a given trait were not observed in the remaining lines. Because of this, the significant finding concerning a L-BMR line on page 5, lines 27-30, is easily missed. Note, that the difference in BMR between the RB and L-BMR lines is even larger than between H-BMR and RB. If results of comparisons of brain size and cognitive abilities of H-BMR vs RB lines are used as a basis for supporting/rejecting particular scenarios of the EB hypothesis, the same logic of reasoning should be applied to the results concerning comparison of the RB with L-BMR

mice. It is not proper that only one side of the results is presented, exposed and used for formulating final conclusions. Why even include the other lines in the research if their results are not mentioned or discussed? The same concerns the lines selected for VO2max.

Our response: We partially disagree. We think there is no need to explicitly remark on all differences, except those directly related to the tested hypotheses. However, on p. 4, lines 13-22 we now report the results of additional analysis of the differences between the H-BMR and L-BMR mice. We have also re-written the last sentence of the section to allude to all line types.

Page 4, Line 15

We suggest omitting this sentence (with the indirect conclusion), which is more appropriate in Discussion. Instead, provide more results per se (as requested in previous comment).

Our response: This sentence briefly presents the major finding of our study. We think it is appropriate to highlight it also in the Results section, particularly since the Reviewers rightly noticed (their comment above) that they are rather complex.

Page 4, line 17

“To compare learning abilities of the line types....” This again is awkward. “Line types” do not have any learning capability (mice from the line types do have). So, you can say “compare learning abilities of mice from the line types,” or perhaps “compare learning abilities among line types (ok, because it does not imply that line types are subjects that learn something). Please consider also in other places where you speak about line types. “Line types” possess statistical properties such as mean values of traits, but not the traits themselves.

Our response: Yes, thank you. Corrected.

Page 4, line 28

Unnecessary comma between “often” and “than.” It should be “VO2max or...,” not “and.”

Our response: We deleted comma. We retained ‘and’ as it refers to all line types other than the H-BMR line type.

Page 4, line 29

Better “random-bred” rather than “randomly bred.”

Our response: Yes, thank you. Corrected.

Page 5, line 3

In comparisons involving more than two selection directions, the correct phrasing should refer to differences “among,” not “between” line types.

Our response: Yes, thank you. Corrected.

Page 5, lines 16-17

The statement “when sucrose solution was replaced with quinine” is imprecise, for it suggests that the quinine test was performed immediately after the sucrose test, by placing a bottle containing quinine solution in the same place where the bottle with sucrose solution used to

be. As we see e.g. on Fig. 2, the two tests were separated by a period when mice were relearning to find water in a cage corner different than the one they used in the sucrose test, and they did not have access to sucrose solution during this time.

Our response: Technically, this description is correct, as explained in the Materials and Methods section (p. 10, lines 8-17). We think that a full repetition of the description of the sequence of trials, suggested by the Referees would be confusing here, as it does not bear a direct relevance to the results.

Page 5, lines 24-30

The phrase “we investigated the differences in the between-line type learning abilities” sounds a bit confusing. Can be rephrased to “we investigated the differences in learning abilities among line types.”

Our response: corrected.

More importantly:

- As in most of the lines in fact no “extinction” was observed (the percentage of freezing increased in time), is the “extinction” (decrease in time) still a valid measure of the learning capability?

Our response: We relegate our response to similar query below.

- The statistical analysis used in this case – repeated measures ANOVA with the 6 time periods treated as categories – is not adequate to test the hypothesis that the extinction rates (slope of change in time) differ systematically in time. You could have the F test with 15 numerator dfs for the line type × time interaction significant even if there are no systematic difference in the “extinction.” The proper analysis could be based on the repeated measures approach, but with the categorical time factor decomposed to represent separate linear, quadratic, etc. trends, or, alternatively, an ANCOVA with time (and perhaps its square to account for nonlinear trends) as quantitative predictor.

Our response: We are somewhat perplexed by this comment. We think that the structure of our repeated measures ANOVA is appropriate. Indeed, the dfs of the nominator of F test was equal 15, which follows from the structure of statistical model. However, to conform to Reviewers’ suggestion we have re-analysed the data with ANCOVA model with the F test with df=3 for the line x time interaction. This interaction was still significant (P=0.015). Thus, we are re-assured that the opposite trend of fear extinction of the L-BMR mice clearly discernible from animals of other line types (Figure 3) is robust. We therefore elected to retain the repeated measures ANOVA model in the paper.

As for possible non-linearity of the fear extinction rate: yes, we tested the significance of the square term. It was not significant.

Page 6, lines 3-6

What do you mean by treating one of the random-bred lines as an “outgroup”? The same appears in Methods (page 13 line 1), but again it is not clear what you mean. Such a term has a special meaning in hierarchical classification analysis (e.g. phylogenetics), but it is not generally used in experimental design context. Perhaps the authors did not want to use the term “reference group” because they realize that the RB lines from the other selection

experiment are methodologically not quite a proper control in the BMR selection experiment. However, applying a vague term is not a satisfactory solution. As we said in general comments, methodological limitations should be openly stated in Discussion and considered as circumstances weakening the conclusions.

At any rate, the way in which you presented results of the analysis in this section does not indicate it is treated as a reference of any kind. Moreover, the statistical result you provided ($F_{2,24} = 18.4$) is not a proper result supporting the claim that “H-BMR mice manifested significantly increased neuronal plasticity,” because this test tells only that there are differences among three groups. Finally, in this analysis no correction for random effects of replicate lines could have been applied, even if in the very paragraph describing this analysis in Methods section you claim that you applied “mixed model.” Thus, what is the argument that the difference in neuronal plasticity is not an effect of a random drift? It is also not clear why animals from only one of the RB lines were taken, instead of a smaller representation of each of the replicate lines. Certainly, there must be differences among the replicate lines in several traits, perhaps including also BMR. Perhaps BMR (or another relevant trait) in this particular line was nearly as high as in the H-BMR mice, or nearly as low as in the L-BMR mice? It is obvious that this would affect the interpretation of the results. So, if you associate the differences in LPT between the groups with differences in BMR, the BMR measured in this particular RB line should be shown for comparison, or at least the text should inform that BMR in this line was nearly the same as the mean from all RB lines (shown on Fig. 1).

Our response: Yes, we admit that for the reasons identified by the Referees we had a problem with naming the random bred subline used in the LTP measurements. Upon re-consideration, we have now re-named it as the ‘reference group’. We realize though, that this does not address the major concerns of the Referees. We agree that proper way of designing the LTP measurements should involve the use of animals from all random bred replicate sublines, even though it still does not fully solve the problem of the lack of a direct common ancestry of the random bred and BMR-selected lines (see our comments on this issues below/above). Due to technical complexity and labor-intensity of the LTP measurements we were simply unable to carry them out on a larger set of animals. We have therefore randomly selected one of the replicate lines as the reference. A comparison of body mass corrected BMR of mice from this line with the animals from the H-BMR and L-BMR line types revealed that BMR of the reference group fell between BMRs of the selected lines (ANCOVA, the effect of line: $F_{2,155}=337.9$, $p<0.001$. Body mass corrected BMRs of mice under comparison averaged (LSM): 64.6 ± 0.7 , 52.8 ± 1.8 , 39.6 ± 0.7 for the H-BMR, reference group and L-BMR mice, respectively. Thus, this pattern of differences matches that found for a full comparison of BMR presented in Fig. 1A. We now report the above information in Supplementary Materials (lines 107-116).

We also agree that the high statistical significance of the difference in LTP between the H-BMR and L-BMR lines (BTW, tested by the Tukey test, see caption of Figure 4) by itself does not exclude a possible effect of random drift. To remedy this problem we have evaluated the between line type difference in LTP of the H-BMR and L-BMR mice using guidelines elaborated by Henderson (1989, 1997) and Konarzewski et al. (2005). Briefly, we first calculated within-line means. We then standardized the difference between those means by dividing it by its SD. Finally, we compared thus standardized difference with theoretical threshold value of its confidence interval expected due to genetic drift, that is, in the absence of genetic correlation between the primary selected trait (BMR) and the LTP. The between line difference in LTP exceeded the threshold value of CI (Figure S1), which strongly

suggests that the pattern depicted in Fig. 4 reflects a genuine correlated effect of selection for BMR.

We present a full description of the above analysis in the section ‘Evaluation of the effect of genetic drift of Supplementary Materials. We refer to this obtained results in lines p. 6, lines 10-13 of the main text.

Page 6, lines 10-11

The word “sinks” is unnecessary (the word is not used anywhere else in the text), and it forces the next part of the sentence to be put in brackets, which were then closed with a precarious “(2)” (mistyped literature reference?). Also, better say “other physiological functions” not “traits” (“trait” means a characteristic or a quality – but energy/resources are allocated to structures or functions).

Our response: Thank you, corrected.

Page 6, Line 11

We would suggest removing from the description of scenario (i) the explicit example of immunocompetence. It is not tested and discussed further, and in the next sentence (Lines 13-14) you say that you “comprehensively tested (i) and (ii),” which suggests that immunocompetence was included in the experiment.

Our response: Thank you, done.

Page 6, line 16

Missing comma before “however.” Generally, punctuation should be checked and corrected in the entire text.

Our response: Thank you, done.

Page 7, line 3

Unnecessary comma between “revealed” and “that.”

Our response: Thank you, deleted.

Page 7, line 6

Missing comma before “which.”

Our response: Thank you, inserted.

Page 7, line 8

This sentence, with the whole long phrase “The above discussed between line-type differences” as the subject, is difficult to follow (to make it clearer you would also need a dash linking “between” with “line”). It sounds better in the form “The above discussed differences among line types...”

Our response: Thank you, corrected.

Page 7, lines 9-10

“Our study is the first...” - You have already said that in Introduction and in the first paragraph of Discussion, and repeating it does not help. Better omit it, especially considering that several parts are written using mental shortcuts, evidently in an attempt to shorten the text.

Our response: Thank you, omitted.

Page 7, line 11

What is a “reinforced bottle”? The term is not used in the methods description.

Our response: Thank you, we have rephrased the sentence to omit this jargon notation.

Page 7, lines 12-13

In the “high- BMR mice” phase there should be no space before “BMR” (“high-BMR mice”). In the “low BMR mice” there should be dash instead of space (“low-BMR mice”). The same concerns also all other places in the text. Generally, the usage of the line type identifiers is not consistent (in other places you use H-BMR, L-BMR).

Our response: Thank you, corrected.

Page 7, line 15

“using the random bred mice as the outgroup.” In this particular sentence the information is in addition misleading because by “random bred mice” you generally mean animals from four replicate lines (as declared in Methods-Animals section), whereas in this specific measurements animals from only one of the RB lines were used.

Our response: We agree. We have now rewritten the sentence to conform to Reviewers’ remark (p. 6, line 8).

Page 7, lines 19-20

Here you speak about the increased value in H-BMR and decreased in L-BMR mice, but in Results you talked only about the result for H-BMR mice. This is only one of the examples showing how the presentation of Results is biased towards results for the H-BMR line.

Our response: We have extended the results section and incorporated additional analyses comparing the H-BMR and L-BMR line types. Please also see our response to similar comment above.

Page 7, lines 25-26

Selection acting specifically on increased overall energy intake does not seem likely, unless it was driven by competition for resources (eat everything before others can get to it), or a demand to gain body mass either to shorten the time needed to grow to full adult size, or to accumulate resources. Either way, selection for increased energy intake would be a byproduct of selection acting on a trait more closely associated with Darwinian-sense fitness. Moreover, the model used in this study presents a scenario with causality inverted with respect to what the sentence states: here, it was selection for high BMR that affected energy intake and gut size. Results from selection experiments are generally difficult to interpret as arguments for hypotheses conceding the order in which selection factors acted in nature. At any rate, this experiment certainly does not warrant the conclusion that “initial selection for increased overall energy intake” was acting.

Our response: We partially disagree. There are good evidences for ‘metabolic acceleration’ that is, selection acting on increased overall energy intake in hominids (Ponzer et al. 2016, cited in our paper). Sure enough, this was not selection on energy expenditures per se, but rather the selection on life history traits was fueled by increased energy intake. Nevertheless, the changes in anatomical and physiological proxies (such as BMR, organs size etc.) of the selection on life history traits remain a strong evolutionary argument in favour of the evolutionary scenario postulated in the concluding paragraph of Discussion. We agree that the sequence of events of this scenario is speculative. Nevertheless, its likelihood is supported by the existence of genetic correlations between key components of this scenario (between BMR and energy intake; BMR and mass of internal organs). We have now highlighted it on p. 7 line 34 and p. 8 in lines 1-2, and added citation of the recent paper by Kozłowski et al. (2020) discussing the issue of BMR being a target of natural selection.

Page 7, lines 27-29

The suggested scenario is alluring, but it lacks explanation of how exactly would neurons become more efficient, and how would the increased BMR, gut size and energy intake support their efficiency. One may assume the increased body temperature, circulation, more thorough digestion allowing access and absorption of otherwise non-digested nutrients could possibly create a more favorable environment for neurons to thrive, but it is not stated in a text.

Our response: This comment has helped us to realize that we have mistakenly referred to neuronal efficiency, instead of neuronal plasticity, that we studied by means of the LTP analysis. We have re-written the sentence that now pertains to neuronal plasticity.

Page 8 and further – Methods section

No information about the number and age of animals used in the experiment and particular tests is provided.

Our response: Age is reported on. p. 8, line 9, the number of animals in Table S1 of Supplementary Materials.

Page 8, lines 1-17

The “Animals” section provides only very limited information about the animal maintenance conditions, which is not only formally required, but also quite important from the perspective of the behavioral tests applied in this work. For example, what was the size of the regular housing cages and how many mice per cage were normally maintained (how does it compare to the 10-11 in the IntelliCages?). What was the photoperiod under regular maintenance (the information is only for the test in IntelliCages). The information about the populations/lines and the method of recruiting the animals for this project is not adequate, either. What is the size of populations/lines (typical number of individuals in a generation), what was the breeding scheme (and hence typical effective population size), and accumulated inbreeding? Perhaps we can assume that the lines are already practically inbred? Has the selection limit been achieved or the selection still progresses? Were the individuals used for the tests randomly sampled from all individuals in the lines, or from a biased group of animals not selected within the regular breeding in the selection experiment? If the latter, has the bias been somehow compensated? Were the animals sampled for a particular experiment unrelated - each from a different family (not siblings or parent-offspring)? If not, how the independence was handled in statistical analysis? Were the animals chosen for the experiment used in any other trials?

Our response: Below, in an itemized fashion, we provide or extend details requested by the Reviewers.

‘...what was the size of the regular housing cages and how many mice per cage were normally maintained.?’

Our response: for information see line 12 of the Supplementary Materials.

What was the photoperiod under regular maintenance (the information is only for the test in IntelliCages).

Our response: for information see line 12 of the Supplementary Materials.

The information about the populations/lines and the method of recruiting the animals for this project is not adequate, either.

Our response: for information see line 15-20 of the Supplementary Materials.

What is the size of populations/lines (typical number of individuals in a generation), what was the breeding scheme (and hence typical effective population size), and accumulated inbreeding? Perhaps we can assume that the lines are already practically inbred?

Our response: We provided this information in ‘Evaluation of the effect of genetic drift’ of Supplementary Materials. In short, the coefficient of inbreeding for selection on BMR averaged 0.3 for generations used in our study. Therefore, the mice are not highly inbred.

Has the selection limit been achieved or the selection still progresses?

Our response: Selection still slowly progresses (particularly in the L-BMR line type), as depicted in figure below.

However, the between generation changes are slow, and for this reason, the effect of generation was not significant in our analyses. The above figure will be published in a companion paper, and for this reason we cannot present them in the current submission.

Were the individuals used for the tests randomly sampled from all individuals in the lines, or from a biased group of animals not selected within the regular breeding in the selection experiment? If the latter, has the bias been somehow compensated?

Our response: for information see line 17-18 of the Supplementary Materials.

Were the animals sampled for a particular experiment unrelated - each from a different family (not siblings or parent-offspring)? If not, how the independence was handled in statistical analysis? Were the animals chosen for the experiment used in any other trials?

Our response: for information see line 15-20 of the Supplementary Materials.

Page 8, line 6

Instead of claiming that the difference between H-BMR and L-BMR lines is just “sufficiently large,” it would be good if you mentioned the actual scope of the difference in the generations from which the mice were sampled, and provide a brief rationale of the claim with reference to adequate literature. You support the claim by reference to Sadowska et al. 2017 paper, which concerns the VO₂max experiment, not BMR-selected lines. Moreover, even though you do have a support concerning the significance of the direct response to selection (difference in BMR), it does not directly apply to differences in correlated responses. Thus, this issue should be somehow approached in the statistical analysis section and in Discussion.

Our response: Indeed, Sadowska et al. is not most appropriate reference. We have replaced it with Sadowska et al. 2019, where detailed analysis of the between-line difference for BMR in generation F50 is reported in Materials and Methods section, quote:

‘Because our selection line types are not replicated, there is always a possibility that the between-line type differences in the traits of interest may be due to genetic drift rather than a genuine effect of artificial selection. For this reason we also analysed the differences in BMR according to Henderson’s guidelines (for details see Konarzewski 2005, Sadowska et al 2015; $h^2 = 0.4$ for BMR; $F = 0.3$; for the effective population size for generation 50). Between line type differences in BMR in the females from the 50. generation were highly significant ($F_{1,61} = 578.11$; $P < 0.001$; H-BMR line type: $69.11 \pm 1.83 \text{ ml O}_2 \text{ h}^{-1}$; L-BMR line type: $40.77 \pm 1.69 \text{ O}_2 \text{ h}^{-1}$). The differences in BMR did exceed those expected to arise from genetic drift alone (the difference (d) expressed as a multiple of phenotypic SD, $d = 7.82$ was higher than that expected under genetic drift, $d_{\text{drift}} = 1.47$).’

We have carried out detailed analysis of correlated responses to selection as described in section ‘Evaluation of the effect of genetic drift’ (Supplementary Materials).

Page 8, lines 8-14

“the Peak Metabolic Rate (PMR) line type” – you use this name of the trait and the PMR acronym only in this place of Methods and in Fig. 1 – but not in the Introduction or Results (where you speak about “maximum aerobic metabolism”). Please be consistent. You have not provided any information about difference in VO₂max in this selection experiment, or information about any relevant correlated responses (e.g. food consumption). And again in the description of the lines you refer to your own, but irrelevant publication (This is a general problem in this manuscript, which undermines credibility of the work!).

Our response: PMR selection is now appropriately referenced. The use of this selection has not been particularly illuminating, so there is no reason to devote too much precious space to describe it. We now cite two papers presenting this selection: Gębczyński and Konarzewski 2009, describing the selection regimen applied and Sadowska et al. 2017, reporting the between line type divergence in the selected trait ($F_{1,6} = 78.65$; $P < 0.001$; body mass-corrected PMR of selected mice: $323.1 \pm 3.3 \text{ ml O}_2/\text{h}$; PMR of Random Bred mice: $247.2 \pm 3.2 \text{ ml O}_2/\text{h}$) and the lack of correlated response in energy assimilation rate.

Page 8, line 13

The term “randomly bred” suggests lack of control over the breeding scheme. If the mice were bred in a planned, but randomized manner, the more appropriate term would be “random-bred.”

Our response: corrected.

Page 8, Measurements of Basal Metabolic Rate

The measurement is not clearly described, and you do not refer to any of your previous papers with a more precise description. You do not even provide such fundamental information as the duration, light phase and timing of the BMR measurements. The text implies that in each of the two systems you had three channels, but it is not clear if you measured simultaneously 4 individuals (2 individuals and reference channel in each of the two systems) or in 5 (one common reference channel for the two systems). What was the schedule of the channels multiplexing, how frequent was zeroing etc.? Please also describe the method of randomization of animals from different experimental groups, and the procedure preceding the measurements – how long animals fasted, did you measure their body mass before fasting, or before BMR measurements etc. Did the animals represent all generations mentioned in the previous section?

Our response: Indeed, we did not refer in this context to the paper by Książek et al. (2004), which reports detailed description of the measurements of BMR. We now direct interested readers to this paper. For the sake of brevity, we have shortened the description (now presented in Supplementary Materials), as it repeats several details reported in Książek et al. However, we have also added information requested by the referees, along with few other important technicalities.

We used separate groups of animals for BMR measurements and other trials. We have now specified it in lines 17-18 of Supplementary Materials.

Page 8, line 31

“BMR was calculated Withers’ equation” - missing preposition. Also, the equation is for calculating the rate of oxygen consumption in general (not BMR), whereas BMR is defined as “the lowest”

Our response: Corrected.

Page 9 – measuring cognitive abilities

For subsequent tests separate “batches of naïve animals” were taken. But how the batches were related to the generations? The “Animals” section informs that females from 4 generations of one selection experiment, and 5 generations of another, were used. Did each batch (and also type of behavioral test) include animals from each of the generations? If not, this is important information, because joint interpretation of results from the set of the tests is not so obvious if each test was performed on different subset of generations.

Our response: We are well aware that ideally, all measurements should have been carried out on the same animals, within the same generation. This is however simply unfeasible because of logistic limitations stemming from, e.g., an obvious requirement of the use of naïve animals in behavioural tests. In several cases we have been therefore forced to use animals of subsequent generations, as indicated in a data file deposited in DRYAD and Table S1. In measurements involving animals from more than one generation we coded generation as a fixed factor in initial statistical models. However, it was dropped in the final analyses, because it was not significant or not estimable. In the latter case we continued the analyses within a given type line type represented by more than one generation. Again, such restricted analyses revealed the lack of statistical significance of the effect of generation.

By ‘batches’ we mean groups of animals that were simultaneously subject to behavioural trials in the Intellicage setups. The batches were coded as a fixed effect. Since this effect was tangential to the main tested hypotheses, we elected to report it in the footnote of the Table 1, rather than the main body of the Table.

We have now provided detailed information on handling the effect of generation and batches in Supplementary Materials (lines 97-101).

Page 9, Line 19

Please provide more details about the procedure of sedation and microtransponder injection, or a reference to the protocol you used. This description is not sufficient to allow replication studies.

Our response: Information on sedation and microtransponder injection is provided on p. 9, in lines 15-16 of the main text. We have added a reference to the paper reporting details of the procedure.

Page 9, lines 22-24

What was the scheme of assigning individuals to particular cages? Does it mean that in one cage individuals from all type of lines were represented? Can the observations made on animals tested simultaneously in the same cage be treated as statistically independent? Had any attempt been made to somehow check it?

Our response: We carried out trials on batches of same line type animals housed in Intellicage system, as indicated on p. 9 in lines 20-12 of the main text. We used individuals from the same line type to minimise possible effect of social context (Kiryk et al. 2011, cited therein).

The effect of batch (i.e. the Intellicage affiliation, as indicated in the data file deposited in Dryad) was entered as a fixed factor and controlled for in the statistical analyses. It was significant in several analyses, as indicated in Table 2.

Kiryk, G. Mochol, R.K. Filipkowski, M. Wawrzyniak, V. Lioudyno, E. Knapska, T. Gorkiewicz, M. Balcerzyk, S. Leski, F.V. Leuven, H.P. Lipp, D.K. Wojcik, L. Kaczmarek Cognitive abilities of Alzheimer’s disease transgenic mice are modulated by social context and circadian rhythm *Curr. Alzheimer Res.*, 8 (8) (2011), pp. 883-892.

Page 10, lines 12-21

There are two aspects of the subject of “correct response” in aversive learning task that makes it rather confusing. First, between the sucrose and quinine test the mice are taught to find water in a different corner of a cage, therefore it is not obvious that they would still expect to find sucrose solution in one of the bottles presented – this is a different corner with a different set of bottles, after all. Therefore, the statement that “the bottle preferred during the previous phase was replaced” refers to the cage corner the mouse has no longer access to. Second, in my understanding the meaning of “correct response” differs depending on the type of solution presented – if sucrose solution is available, it is correct for the mouse to prefer it, but where quinine is presented, it is correct for the mouse to avoid it.

Our response: Indeed, the description was confusing. We have replaced ‘previous phase’ with ‘previous 2 days’ (now p. 10 line 14), which should solve the problem. We have also replaced ‘correct’ with ‘incorrect’. Indeed, in the aversive task a choice of the bottle with quinine is incorrect.

Page 10-11, Fear condition procedure

Was the freezing behavior observed and measured inside the training chamber, but before applying the electric stimuli? The animals could have different “baseline” stress response to placing in such chambers, and distinct characteristic of the changes of freezing percentage in time (different “extinction” rate), irrespective of any electric stimuli, and these characteristics could differ among line types. So, what is the proof that the differences in the “freezing extinction” indeed inform about the intended trait (contextual fear conditioning), rather than just differences in reaction to placing in any chamber, novel or not?

Another doubt concerns using the rapid extinction of the freezing percentage as indication of low cognitive abilities. Such a result could mean that the animal quickly – and correctly! – grasps that the cage should not be feared. Why the susceptibility to being fooled (keeping

being stressed for false reasons) by mice from all the lines except the L-BMR should be treated as an evidence of their superior CA? We are not experts in such test, but we expect that other readers may have the same concern, and we should be convinced that the test results are not ambiguous – or the possible ambiguity should be addressed in Discussion.

Our response: We are puzzled with this comment, as it runs counter well- established paradigm of fear conditioning (e.g., Curzon et al. 2009) and suggests that a fast reduction of freezing response of the L-BMR mice could be adaptive. Freezing in the natural environment is a life-saving response in rodents, e.g., when a predator is nearby. This may last for seconds to minutes depending on the degree of learning and memorising achieved by the subject. Generally, faster learning and better memorization is associated with slower loss of freezing response, not vice versa.

High sustained freezing is repeatedly reported in studies applying 3-min exposure to the context, as we did in our test. The unusually fast fear extinction in the L-BMR mice cannot be interpreted as adaptive, and indicates their inferior CA.

Peter Curzon, Nathan R. Rustay, and Kaitlin E. Browman. 2009. Cued and Contextual Fear Conditioning for Rodents. In: *Methods of Behavior Analysis in Neuroscience*. 2nd edition. Buccafusco JJ, editor. Boca Raton (FL): CRC Press/Taylor & Francis.

Page 10, line 33

I do not think introducing the ITI abbreviation is necessary, as it is not used anywhere else in the text.

Our response: abbreviation deleted.

Page 12, lines 1-4

The analyses apparently did not include respirometric system or the chamber as additional cofactors. Have you checked these factors in initial models?

Our response: Yes, as indicated on p. 11 in lines 19-21.

Page 12, line 10

You included “batches of animals” as an effect, but the section describing BMR measurements does not mention “batches.” The word is used in the section concerning behavioral tests, in the sense of the whole group used in a test. Were the BMR measurements split in time for the same batches? How are the batches related to generations?

Our response: Our mistake. This sentence referred to the analyses of behavioural trials, not BMR. BMR was analysed within single generations for each line type (See Table S1), so it was identical with the effect of the line type. There were no batches of animals distinguished in BMR measurements. See also our response to the query on ‘batches’ given above.

Page 12, lines 24-25

The “dark vs light phase” certainly should be not treated as a “nested factor,” because it is a clear-case fixed effect. The confusion is enhanced by the header “Time(Phase)” in Table 2, which, according to the common convention of describing linear model (e.g. in SAS) means “time nested in phase,” i.e. reverse to what you say here. It is not quite clear what you mean by “nested within the effect of time coded as a fixed factor,” because you have not explicitly said what you mean by the “time.” Is that the difference between the first vs the second 24-h periods? If this is so, as both of the time periods include both phases, you have a clear

example of factorial design – not nested. Thus the model should include Time, Phase, and Time by Phase interaction. However, I would strongly recommend renaming the factors, because they can be confusing (actually, you could reverse their meaning – phase of experiment and time of day). Obviously, the model should include also line type and all respective interactions – also with the Phase.

Our response: Yes, thank you. We have now re-analysed all behavioural trials with statistical models based on factorial design, as recommended by the Reviewers. This has changed the results quantitatively, but not qualitatively and therefore, has not affected our reasoning. Also, we renamed the variables: former variables ‘Phase’ and ‘Time’ have been renamed to ‘Period’, and ‘Day’, respectively. We have re-written Table 2 content to present new results and new notation.

Page 12, line 34

Compared “between H-BMR line types”?

Our response: this should read ‘between the H-BMR and L-BMR line types’. Thank you, corrected.

Page 13, lines 1-3

The general information about the statistical tool used is enigmatic. The reference directs only to a chapter in manual that describes nearly all statistical procedures in SAS, and does not help clarify a number of ambiguities regarding the method used:

- Which specific procedure was used? What you mean by “mixed model extension of a general linear model”? Do you mean specifying random effects within the classical GLM procedure, or the MIXED or GLIMMIX procedure?*
- What estimation method was used?*
- What constraints were applied to the variance terms estimation (non-negative? equal variances?).*
- What residual covariance structure was assumed in the repeated measures models? How the adequacy of the choice was tested? This is quite important issue in repeated measures models, especially as complicated as one of yours – with full factorial design of within-individual effects (Time, Phase) and additionally crossed with a between-subject effect (line type), not to mention some cofactors.*

Our response: We have provided all details on statistical methods requested above in the Statistical Analyses section of Supplementary Materials.

- Did you check the assumption of the homogeneity of slopes in ANCOVA models (e.g. for the relation between BMR and body mass, etc.)?

Our response: Yes, we tested the significance of the line type × body mass interaction. This interaction was never significant ($p > 0.05$) in full models, and therefore was dropped from final analyses. We have now provided information on how we handled interaction on p. 11, line 20-21.

- What tools were used to check basic assumptions of distribution of residuals? Did you have a look at typical diagnostic graphs to at least find out if there were any outliers? Our experience in similar type of data shows that nearly always some outliers appear, so it is strange (if not alarming) that the text speaks nothing about such problems.

Our response: We have now provided requested detailed information on data handling in section Data Handling of the Supplementary Materials.

- A related issue concerns exclusions from some trials due to technical or other failures. Do you really mean that all animals recruited to all the measurements completed successfully the tests and all results were included in final analyses? The lack of the information is particularly suspicious considering that you have not provided information about the number of the animals tested, either.

Our response: As above, this information is provided in the Supplementary Materials (line 69-76) and a data set deposited in DRYAD.

- And also related to this issue is a lack of any information about the results at the individual level – only means for line types are presented.

Our response: New Figure S2 presents individual data points for BMR and organ masses. However, behavioural data and the LTP measurements are massive and comprise up to 800 individual measurements. It is plainly impossible to present them in an illustrative, yet comprehensive manner. We have therefore elected to use least square means from the respective statistical models. Needless to say, anyone interested in a detailed analysis of the original data points can access them from the DRYAD repository.

- We suggest to include more comprehensive results in an appendix, which should show the code used in SAS analyses and extensive outputs from SAS procedures, and relevant graphs showing individual data points. Note that several of your analyses are ANCOVAs, but you do not show any information about the relation between the response variables and the quantitative predictors on X-Y plots, which can lead to gross confusion.

Our response: We report the significance of all key interactions throughout the main text. We have now presented SAS codes in the Supplementary Materials. As for individual data points, please see our response above.

Figure 1

In bar plots the Y axis has to start with a 0, otherwise you risk exaggerating the differences between the groups. You may consider either breaking the axis or presenting your data as points with whiskers, a plot type not restricted by axis scale.

Our response: we disagree. If the differences are exaggerated, then the respective SEs are exaggerated too, so visual proportions between differences and errors of estimate remain unaffected.

The order of groups presented on graphs is confusing. It is understandable that the leftmost bar represents the random-bred, control group, but the next bar should represent the PMR group as the two come from the same selection experiment. In the present form the graph makes a false impression that the RB represents a methodologically proper control with respect to all other line types, as if they appeared in one experiment, which is not correct. The graphs should rather underlie, not hide, that the results are from two independent selection experiments. Perhaps you should even consider applying a color palette that would indicate the two selection experiments and the direction of selection in the more intuitive manner.

Our response: Yes, we have re-arranged the order of bars. We have highlighted the difference between selection experiments in several places in the paper (e.g., p. 8 lines 12-13, p. 12, lines 20-26). We see no reason to complicate Figure 1 by adding another layer of information on the origin of data from two independent selection experiments.

The whiskers and letters presented on graphs are small and faint, I would suggest increasing the font size and line width. Also, whether to plot mean \pm SE or least squares means \pm 95% CI is a matter of taste or journal's preferences, but for me, as a reader, it is easier to visually compare groups when confidence intervals are provided.

Our response: We stick to least squares means \pm SE. We have changed the font and line style.

At any rate, after a careful look one can see that the graphical information about SE is incompatible with the claim that BMR differs significantly between the RB and PMR line types (mean of one of the groups is within the range of SE of the other, not to speak about CI range). Thus, either the information about significant difference is wrong (for PMR the letter should be "a" not "d"), or the SE shown are not those that are the basis of the post-hoc tests (probably the post-hoc tests were not based on the proper error term or dfs).

Our response: Yes, thank you, corrected.

Figure 2

Please provide the information about the whiskers meaning in the description of B and C figures.

Our response: Done.

Please include also the extension of the acronyms of different experimental groups.

Our response: Acronyms are now fully explained in the caption of Figure 1, with reference to other figures.

Figure 2A: The upper and lower row should be described shortly inside the scheme.

Our response: Unfortunately, we haven't found a way to squeeze an additional description, which would not obscure visualisation. We have therefore decided to leave it as it is.

Figures 2B and 2C: The y-axis can have the same unit range; it will be easier to compare the number of nose pokes in both tests.

Our response: The tests are not directly comparable, as they address different aspects of learning. We would therefore like to retain different scales.

I appreciate the way the experimental design is visualized on the figure. However, I find the blue and green indicators of tap water or sucrose solution difficult to discern. The shapes of points on Figs 2 B and C are also difficult to discern, mostly because of relatively small size and pale colors of point outlines.

Our response: Corrected.

The figure description does not indicate what values are represented by points and whiskers. Moreover, the letters on the graphs are misleading, as they do not refer to differences among the points, but the slopes of lines connecting them. It took me a moment to realize why the H-BMR group on element C is marked with an “a” indicating significant difference from other groups if the error bars of its point are overlapping with two other points.

Our response: Thank you, corrected.

Line 15: What do you mean by “(pairwise a-priori t-test with $df = 6$)”? By all appearances all possible pairwise comparisons were performed. If all possible comparisons are planned, the “planning” is meaningless, and the correction for multiple tests should be applied in the same way as for a posteriori tests.

Our response: Our mistake. We carried out planned custom hypothesis tests by means of the Contrast statement of procedure Mixed. So the correction for multiple tests does not apply here. In the Figure 2 caption and Supplementary Materials (lines 102-104) we have added an information on the use of contrasts.

Figure 3

The figure title is confusing. The values show not “extinction,” but just freezing percentage. Perhaps the “extinction” is supposed to mean “decrease of the percentage in time” – but in that case “changes of extinction” is nonsensical. Moreover, in most of the groups the percentage actually increases in time, so speaking about “extinction” is particularly misleading. The figure description does not indicate what values are represented by points and lines. Why are there no whiskers? The description at page 11, lines 4-5 may suggest that the values were averaged within a line type, but the description is not clear enough.

Our response: We have changed “extinction,” to ‘response’, which should solve the problem. We have also amended the figure and provided requested information in its caption.

Figure 4

The figure description does not indicate what values are represented by points and whiskers.

Our response: Added.

General comment on Table 1 and Table 2

Please inform what the bold font means or remove the formatting. I suggest changing the Table 1 description: do not tell generally about the “anatomical traits” but inform directly “organ masses”.

Our response: Corrected.

General comment on References

You have doubled the first number on the list of references, and therefore all further numbers are incorrect.

Our response: Corrected.

Data in repository

The data available in repository contains only raw data, but not the code used for statistical data analysis and full outputs of results from the analysis, which would be also valuable in the case of such a complicated procedures. The additional information could be in a form of electronic appendixes to the paper.

Our response: In the Supplementary Materials we have now provided SAS codes for main analyses.

*The data in the repository is not complete and the description also needs amendments:
- For BMR data, no information on the animals identity is given, and the ID information is not compatible for all the datasets.*

Our response: Information on generation, batch affiliation etc. is now presented. As indicated in lines 16-19 the Supplementary Materials we used different groups of animals for BMR measurements, anatomic analyses and behavioural trials, so the ID cannot be compatible for all the datasets.

- No information on generation and batch is given and other variables that were or could be used in data analysis: age of the animals, the date and time of the measurements (possibly important both for BMR and organ masses), the identifier of respirometric system/chamber, grouping of the animals in the same IntelliCages, possibly also grouping according to family relations, etc. Thus, the data do not allow to even repeat the analyses the authors performed, not to speak about trying alternative models.

Our response: We have now provided the requested information essential for repeating our analyses. However, we see no justification for providing e.g., the exact date of measurements. Information on age of the animals is provided on p. 8 line 9 of the main text.

- There is no explanation of missing values, denoted by “.”. Were the values not measured incidentally, or excluded because of detecting a technical fault afterwards, or excluded from final dataset as severe outliers? Why, e.g. masses of some organs are missing?

Our response: We have now deleted ‘.’ Throughout the data set. Originally it coded missing data, in accordance with the SAS notation. Please also see our response to similar query above.

Appendix B

REVIEW of resubmitted manuscript: **Brain size, gut size and cognitive abilities: the energy tradeoffs tested in artificial selection experiment**

Comments to authors

General comment

The authors responded adequately to majority of our comments, and the manuscript is now much improved. However, some of the deficiencies, e.g. inconsistent terminology, were corrected only in the places we indicated directed, rather than in the entire manuscript. In a few cases, in which the authors chose to not follow our advice, we do not agree with their arguments, and we now explain the points more explicitly. Most importantly, a few important aspects of the statistical analyses remained not corrected. Unlike in the cases of presentation style or the results interpretation, in which we respect the authors right to stay with their opinion, in these few cases it is a matter of presenting results that are legitimate or not. Importantly, however, even though our list of comments is again extensive, we are confident that the authors can resolve the issues and provide a really valuable contribution, and the next revision of the work could be rated even as excellent. Also importantly, this opinion will not change even if the revised analyses would led to somewhat different conclusions.

Major issues

1. Inconsistencies in the names of factors and variables

In the Methods section concerning “Measurements of cognitive abilities” the authors use the words “phase” when referring to subsequent stages of the behavioral tests. However, in Results, on page 5 line 23 the authors report a test of significance for “line-type \times Day interaction”, and in line 27 of “line-type \times time interaction”. It seems, however, that those are the same interactions – that the “time” means here the same as “Day” (not clear why just “Day” is written in capital letter”), and both describe the difference between the last/first 24 h of the subsequent phases. However, just in the next paragraph of the same subchapter, on page 6 line 2, the authors again report significance of “line type \times time interaction” – but in this case the “time” has completely different meaning: time measured in seconds (but, probably, represented in the statistical model as categorical factor). Even worse, the SAS code for the repeated measures model has “time” and “phase” as the predictors – but in this case “phase” refers to a completely different aspect – difference between light and dark phase of day. It takes some time to figure out that here “time” corresponds to the phase of experiment or “Day”. Finally, in Table 2 the same (?) factors are named Day (which in other places is represented by “time”) and “Period” – and what it means is a real puzzle. According to Methods-statistics section (page 11 line 27), the “Period” represents “a continuous 12 h period of observations”, but in SAS code for the repeated measures model such a variable is not present, and the “phase” has a different meaning – unless the boundary of the experimental phases occurred exactly at the onset of the light or dark phase. Moreover, the title and legend of Table 2 are enigmatic and provide no explanation of the columns, and I could not find information on the timing of switching between the experimental phases.

I hope that it does not need an additional argumentation that such a confusing information makes the communication extremely difficult. Thus, to avoid further confusion, the authors must carefully choose the terminology and use the names of factors and variables consistently in all the text, including tables, SI and the SAS code. Please, do not make corrections just in places we indicated directly, but revise the entire text. We realize that it is difficult considering complexity of the research presented in this report – but even more the special attention should be given to the consistency of the terminology.

2. The usage the repeated-measures model with “time” as categorical factor as a bases of the inferences concerning the slope of the freezing extinction

To our comment concerning this issue (Page 5, lines 24-30 of original version) the authors responded:

We are somewhat perplexed by this comment. We think that the structure of our repeated measures ANOVA is appropriate. Indeed, the dfs of the nominator of F test was equal 15, which follows from the structure of statistical model. However, to conform to Reviewers' suggestion we have re-analysed the data with ANCOVA model with the F test with $df=3$ for the line \times time interaction. This interaction was still significant ($P=0.015$). Thus, we are re-assured that the opposite trend of fear extinction of the L-BMR mice clearly discernible from animals of other line types (Figure 3) is robust. We therefore elected to retain the repeated measures ANOVA model in the paper. As for possible non-linearity of the fear extinction rate: yes, we tested the significance of the square term. It was not significant.

The response indicates that the authors have not recognized the merit of our criticism. We never said that the calculation of the dfs for the overall effect of interaction in the repeated measures model was wrong. The repeated measures structure with “time” treated as categorical factor with 6 groups can be indeed applied, but the F test for the overall effect of “time*line type” (with 15 dfs in numerator) is NOT a legitimate basis for the claim that there the line types differ is a systematic trend with time (i.e. that slopes with respect to time differ between the line types). Imagine a simpler case with only 3 time periods, and the mean value of a response variable Y changing with time as follows: 1, 10, 1. The overall F test (with 2 dfs in numerator) would likely indicate a significant effect of “time”, but it would only mean that there are differences among means of the 3 time categories, and NOT that the Y decreases or increases systematically with time; the slope of such a relation is exactly zero. Suppose also that in a second group the pattern of changes in time would be reversed: 10, 1, 10. The overall test of the group*time interaction (with $2*1=2$ dfs in numerator) would probably indicate that the pattern differs between the groups, but this would be NOT a valid bases for a claim that the slopes of the relation between Y and time differ between the groups – because in both groups the slopes equal exactly zero. For the same reason, in the case of your data, the overall test of line type \times time interaction with 15 numerator dfs is a valid basis for a claim that there is a heterogeneity in the pattern for the four groups, but NOT a valid bases for the claim that the slopes of the systematic trend in time differ among the four groups.

In the response the authors say that they have performed (as we recommended) also ANCOVA with time treated as quantitative predictor and got “the same” results – in the sense that the interaction was also significant. However, the fact that in this case conclusions incidentally

agreed is not a logically valid basis for reporting results from the non-adequate model. The ANCOVA is still not a perfect solution because it assumes that observations performed for a given individual in the subsequent observations can be treated as independent, rather than representing a time series. Thus, a more elegant and robust, but also more complicated approach, would be to perform analysis or polynomial contrasts within the framework of the repeated-measures ANOVA such as you fitted, and test for the interaction between the line-type effect and the first order (linear) component of the five possible orthogonal polynomials. Such an analysis would be more appropriate, because, with a proper residual covariance structure, it would correctly account for non-independence of measurements obtained in the subsequent periods. However, by all appearances, even though the authors specified in the Mixed procedure of SAS the “repeated” statement, in fact they have not fitted a repeated-measures model, because they have specified “variance components” covariance structure (the “type=vc” option in the “repeated” statement). Thus, they actually assumed that the observations at the subsequent time periods can be treated as independent and with equal variances. If preliminary models showed that the assumption is reasonable (i.e., models with more complex covariance structures have not performed better) it also means that the much simpler analysis of covariance with time as the quantitative trait (and ID as random effect) would be also fully legitimate, and much easier to present, because it would directly provide estimates of the desired slopes. Note, that it is still not clear how the parameters of the lines shown on Figure 3 were obtained, because the SAS code you have shown in the supplementary materials certainly has not provided the relevant estimates (actually, the code is shown for a different analysis, but I understand the logic of the model was the same).

Thus, the result of the F test presented on page 6 lines 2-6 (“ $F_{15,30} = 2.27$; $p = 0.03$ ”) would be a valid basis for the statement that it:

“reflected the heterogeneity of the dynamics of fear extinction in the studied line types,”

but is NOT a valid basis for the statement that it:

„accounted for the low-BMR mice losing fear response much faster than other line types.”

Again, it is not a valid basis because you could have the significant interaction even if there is no difference in the overall linear trend, i.e., in the slopes of the regression of the response variable on time would not differ.

In addition, in both cases the overall interaction means only heterogeneity among the four groups, and not that a particular group (low-BMR) differs from the others – for the same reason as in any ANOVA with more than two groups the overall test of the grouping effect is not the same as a test that a particular group differs from others. The authors justify the distinction of H-BMR group by claiming that they have performed “a priori contrasts”, but this claim rises another concern (next point).

3. The application of *a priori* contrasts

In response to our comment to Fig. 2 concerning the issue of using “pairwise a-priori t tests” the authors responded:

“We carried out planned custom hypothesis tests by means of the Contrast statement of procedure Mixed. So the correction for multiple tests does not apply here.”

and in Supplementary Information, page 4, lines 102-104 provided a brief information that: *Planned custom hypothesis tests of the differences in slopes of the changes in the number of nosepokes depicted in Figures 2 and 3 of the main text were carried out by means of the Contrast statement of the procedure Mixed of SAS”*.

This information was also repeated on Fig. 2 and 3 legends.

The response shows a gross misunderstanding, and in fact deepens the doubt concerning the whole results of the statistical analyses. The “contrast” statement performs a test for a contrast without correction for multiplicity, but whether it is legitimate to apply uncorrected contrasts or not does not depend on what command of the statistical procedure you use, but on whether the contrasts were really *a priori* planned, and planned with respecting the rules for such contrasts. The limitation for contrasts not requiring the multiplicity correction is that a) the number of such contrasts to the degrees of freedom, i.e., to three contrasts in the case of four groups, and b) that the contrasts are independent (orthogonal). Thus, the analysis of such contrasts, by definition, cannot lead to results that can be presented in the way the authors did it – by using different letters to show means or slopes in four groups differ from each other (such as used on Figs. 1-3). It is indeed very helpful that the authors showed in SI the SAS code used for the analyses, because it confirmed that the criticism is justified. For the analyses of the traits related to the reward and aversive learning tests, the authors performed 3 contrasts, but the contrasts are not independent (not orthogonal), because in each of the contrast one group is compared with all others. Moreover, the code shows also that in fact the authors performed also multiple paired tests between all groups and subgroups without correction for multiplicity (the statements “LSMEANS line/ tdiff”, “LSMEANS line*time/ tdiff”). Certainly, it is not legitimate within one set of analyses to perform for the same groups both the test for all possible comparisons among pairs of groups and some selected contrasts without corrections for multiplicity. In fact, in this case the correction for the test concerning the line*time should be made for a total of 41 contrasts: 3 performed with the contrast commands and $(8*8-8)/2=38$ by the “lsmeans line*time/tdiff” command (because in this case “time” is a factor with two groups, so there are $4*2=8$ subgroups). But, obviously, the 38 simple comparisons are not relevant – none provides the answer whether the rate of learning differs between the line types - and it is not clear why they were at all requested.

The SAS code reveals also that uncorrected multiple comparisons were performed also in the analysis for the comparison on mean values of BMR and anatomical traits. Thus, the results on Fig. 1 are not proper, either.

Most important, the authors should recognize that they cannot claim that they have performed “planned contrasts” if they never told what the planned contrasts were. The *a priori* tests are legitimate only if the plan was established before performing the research and seeing the results, and the planned contrasts are compatible with the *a priori* hypotheses based on external information. Noting in the text of Introduction or Methods indicates that a set of contrasts was *a priori* planned. The hypotheses presented in Introduction are far too general to be a basis for a valid set of planned contrasts. Note, that the contrasts include all four line types, whereas the random bred (RB) line type is not even mentioned in Introduction. Moreover, even if you could present a specific *a priori* hypothesis for the comparison of the High-BMR vs Low-BMR lines,

nothing in the text of Introduction indicates that you had such hypotheses for the other possible comparisons. Actually, even in the response to our comments you made it clear that only for this comparison you had a strong basis, whereas the other lines were of secondary importance and their inclusion on the analyses had an exploratory character. Therefore, applying the *a priori* contrasts (even if they would be technically correctly specified) was not methodologically valid, and consequently the reported „significance” is not justified - not only for results presented on Fig. 2 and 3, but also for those on Fig. 1.

Obviously, if after all my comments the authors would now present a clear set of hypotheses leading to a set of orthogonal contrasts, those would be not “a priori” hypotheses and contrasts. Therefore, in my opinion the only proper solution is to perform the *post-hoc* comparisons, with appropriate correction for the multiplicity of a given set of tests. Note, that even if the results would force changing some conclusions, i.e., some effects would no longer appear significant, this would not decrease the value of the work. We all agree that “p-hacking” is not the path we want to follow.

4. The logic of the repeated measures model for the results of the reward and aversive learning

According to the description in text (page 10, lines 18-23) the data were automatically summed for 12-hour continuous periods, but the results are reported for the critical 24 h periods (the last day with water and the first day with sugar or quinine). The SAS code in SI shows that actually the response variable was the number of nose pokes in one of the “phases”, but the “phase” refers to photoperiod phase (light or dark) – and the correspondence between continuous 12 h periods and the light/dark phases would be strict only if the change of the state from “water” to “sugar” or “quinine” was made at the onset of light phase or onset of dark phase. Otherwise, one of the phases would actually consist of two parts, split by the other photoperiod phase. The authors do not inform at what hour the bottles were changed, so the reader cannot guess. Splitting the timing would not affect the means for the whole 24h periods, but it would certainly affect the estimate of the effect of the “phase”. This was not a focal aspect, but the problem affects also validity of the specification of the repeated measures model, in which the “repeated” factor was encoded as the “time*phase” interaction (where “time” variable is used, inconsistently with declaration in Methods, as the categorical factor distinguishing two “experimental phases” or Days). The repeated measures factor should describe the real order of obtaining the results. If the timing of the two 24 h periods is not aligned with the light/dark phase, the coding used by the authors does not describe properly the structure of the repeated measures design. In addition, even if it is aligned, but the bottles were replaced at the onset of light phase, the repeated statement used by the authors would not provide the proper order, because by default SAS orders the groups defined by text codes alphabetically, and therefore phase=dark comes before phase=light. Thus, the order of day/night would be reversed (this could be amended by changing the codes of the categories, or requesting a change of reference group). The coding of the repeated measures design would be correct only in the case when bottles were changed at the onset of dark phase. If the bottles were, as I suspect, changed sometime during the day, all the repeated measures model is invalid.

Note that, even though the results of the significance tests for the “phase” factor and “time*phase” interaction are reported in Table 2 (but named there “Period” and “Day”), they are not mentioned at all in Results or Discussion; all the inferences are based only on the test for the

effects of “time” (Day) and “line” (which in the entire test is named “line type”, whereas “subline” in SAS code corresponds to “line” in all the other text...). Also, all the least-squares means you show on figures are values averaged across the two 12 h periods, i.e., represent whole days. Thus, why you have at all bothered with performing and presenting the complicated analysis with the repeated-measures model for 12 h periods, rather than a much simpler model in which the dependent variable is a value averaged across whole 24 h periods? Yes, the effect of phase/period was significant, but this is really a trivial observation that the behavior of mice differs between light and dark phase. Perhaps more interesting are the significant interactions between the phase and line type – but you have not discussed the results, either. Note, that with the simpler model Table 2 would have 3 columns less, and the readers would be not distracted by many F and p values that are never referred to in the text. I do not insist you should make this radical change, but if the switch of light/dark phases is not synchronized with the timing of the switch from water to sucrose or quinine, this would probably be the most straightforward solution to the difficulty of specifying a correct structure of the repeated measures model.

If the authors would like to discuss the above statistical issues, please contact pawel.koteja@uj.edu.pl (one the persons who prepared this review).

Minor points

Page 2 lines 6-8 - We fully understand your decision to use the terms “line type” as used in Garland's group. Note, however, that unselected, random bred lines are not mentioned – neither here nor in the Abstract in general. Perhaps you could add the phrase “, and unselected lines” at the end of this sentence and save three words in other places of Abstract. For example, in lines 11/12 the phrase “to the other line types” is not necessary, because the previous sentence makes it clear to what the superiority refers.

Page 3 line 25 - Again, unselected, random bred line type is not mentioned. Therefore, the first sentence of Results section (page 4. line 9) is confusing, because it speaks about four line types, whereas only three are mentioned in Introduction.

Note, that the exemplary sentence we provided in our previous review included the unselected lines, but you have omitted this part.

Page 3 line 31, page 7 line 7 - PMR, CA and LTP acronyms were introduced or spelled out earlier in the main text

(even though CA and LTP were introduced in the abstract, it should be explained in the main text, too)).

Page 4 line 15 - the acronym H-BMR was spelled out (High-BMR) earlier in the text, but L-BMR was not. However intuitive the annotation may seem to be, such acronyms should be

introduced properly. Perhaps even as early as in page 3 line 25, where the line types are mentioned for the first time.

Page 4 line 22 - I believe your conclusion was meant to state that you did not observe the trade-off and did not find a correlation between brain size and BMR. However, the phrasing of the second part of the sentence, starting with "as well as showed", makes an impression that you did find the brain-BMR correlation. I suggest to rephrase the statement to make the point clearer. Also, by making an emphasis on the personal aspect ("we did not observe, [we] showed") you actually decrease the strength of the results. I think that a simple statement would serve better, e.g.: "... genetic drift showed neither brain-gut trade-off nor positive genetic correlation between BMR and brain size."

Page 5 line 3 and throughout the text - there is an inconsistency in the BMR-selected line type names: sometimes the "High" or "Low" is capitalized and sometimes not, and occasionally it is abbreviated, and sometimes written without the dash between the "high" and "BMR". Note, that we have made the same comment in first review, but some of the inconsistencies remained not amended.

Page 5 lines 24 and 29 - these two statements are contradictory: "when sucrose solution was replaced with quinine" and "once the bottle with tap water was replaced with the bottle containing quinine solution". We pointed out the issue of "replacing" in our previous review, and the revision amended it only partly.

Page 5 line 34 - page 6 line 2 - this sentence is confusing for a first-time reader. Literarily, you wrote that after conditioning phase, you measured the extinction of freezing... "in another group of mice", i.e. not those that were conditioned. The information that another group of mice was used would be better given in the previous sentence (Finally, in another group..., we investigated...). Also, even though this is obvious to specialists, you could say more explicitly that response to "perceived threat" is investigated with no shocks applied ("...response to perceived threat (i.e., in the absence of the electric shocks)").

Page 7 lines 16-17 - the text is not grammatically or logically correct. The rhetoric "both" requires "and" linking some two objects or statements. It could be, e.g., "Both, learning the position of bottle with sucrose and quinine in the tests performed in the IntelliCage, involve hippocampus ." However, as this is Discussion, perhaps it would be better to make a more general statement, such as "Both aversive and positive learning, such as in the tests we performed in IntelliCages, involve hippocampus" (but this is surely up to the authors).

Page 7 line 28 - the abbreviated label of the random-bred line type (RB) was not introduced in the text. Again, I suggest introducing all line types and relevant abbreviations already in the Introduction, which will help to use the abbreviations consistently.

Page 8 lines 10-11 and 17 - misleading mental shortcut: divergent selection is applied to mouse lines, not to mice. Saying that mice were selected suggests that animals expressing a particular trait were picked from a general population for the purpose of a given test (as in "we selected patients with the highest / lowest heart rate for a cardiac drug study"). Selection applied on lines implies multigenerational, evolutionary design. We realize that the shortcut form is fully understandable among researchers familiar with experimental evolution, but our own experience shows that many physiologists and neurobiologists from biomedical community have a problem with proper understanding the selection experiments, and therefore it is better to be explicit. Also, the phrase "for high/low BMR" literarily means that you selected for a ratio of high to low BMR. The sentence should rewrite as: "we maintain lines of mice divergently selected for high or low body-mass-corrected Basal Metabolic Rate (BMR), quantified..."

Page 9 line 17/18 – "transponder *implantation* procedure."

Page 9 line 22 - the statement regarding "excluding" competition for access to the bottles seems to be too strong. There were less bottles than there were mice, and during the learning and test phase each mouse had to share access to their only source of water (a single cage corner) with one or two other mice. In such situation competition cannot be fully excluded, but it can be reduced/limited/minimized, as you phrase it on page 10 line 2.

Page 9 line 29 - "pokes" should be in past tense.

Page 10, line 11 - "During the next phase lasting 2 days," – either comma after „phase" (as in line 13), or in this case better: „During the next 2-day phase".

Page 10 lines 13-14 - I do not understand "the bottle preferred during the previous two days". During the previous two days both bottles contained tap water. It can be imagined that a mouse may prefer to drink water from one of the two bottles it is offered, but the or two other mice assigned to the same corner (page 10 line 1) can vary in their spatial preferences. Hence, placing quinine solution in place of a water bottle preferred by one of the mice would likely mean replacing the less-preferred water bottle of another mouse.

Page 12 line 15 - "drawn", not "drown"

Page 17, Figure 1 legend - repeated information in lines 8-9 and 13-15. In line 9, before "RB-" there should be " , and" rather than period.

Figure 1 – to our comment on previous version, in which we pointed out that the bar graphs should begin with zero on Y axis, the authors responded:

“we disagree. If the differences are exaggerated, then the respective SEs are exaggerated too, so visual proportions between differences and errors of estimate remain unaffected.”

The response is not adequate. Indeed, the proportion of the difference to error is represented correctly, but in the case of bar graphs our brains tend to first of all spot the ratios of the heights of the bars for different groups. Thus, when we look at Fig. 1a, the first thing we notice is that the H-BMR bar is twice as high as the L-BMR one, and only after reading the values on the Y axis we can realize that the gap between 40 and 80 indicates that the first assumption cannot be true - but it is impossible to tell by how much, because there is no label at the intersection of the x and y axes. This is not only an inefficient, but also a misleading way of presenting the data. The rule that bar charts should start from zero is widely recognized by experts of scientific communication. If you do not believe, just google “bar chart starts zero”, and in front of the long list of resources you can get, e.g.:

<http://www.chadskelton.com/2018/06/bar-charts-should-always-start-at-zero.html>

where the first sentence says: “If there's one thing almost everyone agrees on in data visualization, it's that bar charts should start at zero.”

As the authors are from Poland, you can consult also the guide of January Weiner “Technika pisania i prezentowania...”, which some of the authors surely know.

Figure 1a - Y axis unit is “ml zero-squared per hour”?

Fig. 1b brain - I understand that there were no differences in brain mass among types of selection. Why bother putting all the letters “abcd”? Why not just “a” above each?

Figure 2 description, line 2 and 6 - "row" not "raw".

Figure 2 legend line 13 and 15 – “least square means” - it should be “squares”.

Figure 2 - again, there is no reason to use multiple letters “cd” on Fig. 2b and “bcd” of Fig. 2c. More importantly, to our suggestion that Y axes on Figs. 2Ba 2C should have the same scale, the authors responded that:

“The tests are not directly comparable, as they address different aspects of learning. We would therefore like to retain different scales.”

In our opinion the argument is not satisfactory. Even though different aspect of learning is assessed, the same behavior is directly measured, and in both tests the starting condition is the same (“water”). Therefore, it is meaningful to ask whether the behavior of the animals at the start was the same. The different scales on the Y axes obscure the fact that the number of nosepokes with pure water was only about 15-20 in the first test, but about 25-38 in the second test. The

difference was in fact of the same order as that between water and sugar. The authors have not tried to explain the difference, which – with the figure properly prepared – would be more striking and would certainly raise a question about the explanation. One possible explanation is that the test was performed on a different group of animals. However, if we correctly understand the description of the scheme, the results presented on Fig. 2B (test with sugar) are pooled results from the two groups of animals – correct? If this is the case, did the two groups differ in the first test, and if yes, what could be the reason? Another possibility is that passing the first test modifies behavior of the mice so that they behave in a different way also under exactly the same condition (in the phase with water). If this is the case, however, the results of the second test – of the aversive learning – cannot be treated as independent of the first one. In other words, can we be sure that the results of the aversive learning would be the same if the test were performed on really naïve animals, rather than those that had passed through the reward learning test? Thus, we suggest to not only amend the presentation (show the same scale on Y axes), but also to discuss shortly both the difference in the results with water, and the issue of non-independence of the two learning tests.

Figure 3 legend (and methods regarding this measurement) - still not sure what was measured, what does the percent represent. Was the behavior scored second by second, and a fraction of immobile seconds within each 30s period was calculated per mouse? Or were the mice given a 0 or 1 score depending on whether a particular 30-s interval featured at least one incident of freezing, the percent representing a line mean? This should be clarified better in the text.

Table 2 line 6 - this is the first place in the text where the word “batch” appears. The effect was not mentioned in the Results or Methods sections. In the SI file in line 98 the word appears, as one of the effects included in model, but with no explanation of what the “batch” means. We can only guess that “batch” distinguish between the animals that were used in only the reward learning and those that passed both tests. The significant effect of batch can provide a clue concerning my earlier inquiry concerning the difference in number of nose pokes during the phase with pure water (Fig. 2).

Suppl., line 1 - missed one "p" in Supplementary."

Suppl., line 42 – even though LTP is explained in the main text, it would be good to be explained also in the first occurrence in the Supplement.

Suppl., line 109 - what was randomly selected - animals, replicate line, or both? It can be said better in a form of:

"We have therefore restricted these measurements to mice from only three lines: H-BMR, L-BMR and one randomly selected replicate line belonging to the RB line type, which was treated as a reference."

Figure S1 - the blue line of the confidence interval looks like just another plotted variable, which for some reason was not introduced in the figure legend. The distinction would be clearer if the line was drawn in a different style, e.g. without symbols

Table S1 - I appreciate inclusion of such a detailed table, but one fragment is puzzling. According to Methods section the sucrose preference test was performed on two groups of mice, with the test on the second group extended by a quinine avoidance test. However, the generation numbers are completely different for the rows concerning the test with sucrose and quinine. Thus, this would suggest that, even though on the second group the test with sucrose was performed, the results were not used in the data analyses. Is that correct? But if this is the case, what is the “batch” effect in the ANCOVA model? Also, should not the row for quinine be labelled “incorrect” (as in other tables)?

An extra column presenting a total number of animals in each group would also be appreciated, even if it is not entirely necessary for proper comprehension.

Appendix C

Response to the Referee's comments on

Brain size, gut size and cognitive abilities: the energy trade-offs tested in artificial selection experiment

by

Anna Goncerzewicz, Tomasz Górkiewicz, Jakub M. Dzik, Joanna Jędrzejewska-Szmek, Ewelina Knapska
and Marek Konarzewski

We thank the Reviewer for his in-depth comments. To assist the Referee, we provide a clean amended copy of our submission along with a copy with parts of the text (marked in green) affected by the revision. In our responses to the Referee's comments given below we refer in an itemised fashion to specific lines of the text.

Comments to authors

General comment

The authors responded adequately to majority of our comments, and the manuscript is now much improved. However, some of the deficiencies, e.g. inconsistent terminology, were corrected only in the places we indicated directed, rather than in the entire manuscript. In a few cases, in which the authors chose to not follow our advice, we do not agree with their arguments, and we now explain the points more explicitly. Most importantly, a few important aspects of the statistical analyses remained not corrected. Unlike in the cases of presentation style or the results interpretation, in which we respect the authors right to stay with their opinion, in these few cases it is a matter of presenting results that are legitimate or not. Importantly, however, even though our list of comments is again extensive, we are confident that the authors can resolve the issues and provide a really valuable contribution, and the next revision of the work could be rated even as excellent. Also importantly, this opinion will not change even if the revised analyses would led to somewhat different conclusions.

Our response: We appreciate inquisitiveness of the referee, who pinpointed several deficiencies of our presentation.

Major issues

1. Inconsistencies in the names of factors and variables

In the Methods section concerning "Measurements of cognitive abilities" the authors use the words "phase" when referring to subsequent stages of the behavioral tests. However, in Results, on page 5 line 23 the authors report a test of significance for "line-type × Day interaction", and in line 27 of "line-type × time interaction". It seems, however, that those are the same interactions – that the "time" means here the same as "Day" (not clear why just "Day" is written in capital letter), and both describe the difference between the last/first 24 h of the subsequent phases. However, just in the next paragraph of the same subchapter, on page 6 line 2, the authors again report significance of "line type × time interaction" – but in this case the "time" has completely different meaning: time measured in seconds (but, probably, represented in the statistical model as categorical factor). Even worse, the SAS code for the repeated measures model has "time" and "phase" as the predictors – but in this case "phase" refers to a completely different aspect – difference between light and dark phase of day. It takes some time to figure out

that here “time” corresponds to the phase of experiment or “Day”. Finally, in Table 2 the same (?) factors are named Day (which in other places is represented by “time”) and “Period” – and what it means is a real puzzle. According to Methods-statistics section (page 11 line 27), the “Period” represents “a continuous 12 h period of observations”, but in SAS code for the repeated measures model such a variable is not present, and the “phase” has a different meaning – unless the boundary of the experimental phases occurred exactly at the onset of the light or dark phase. Moreover, the title and legend of Table 2 are enigmatic and provide no explanation of the columns, and I could not find information on the timing of switching between the experimental phases.

I hope that it does not need an additional argumentation that such a confusing information makes the communication extremely difficult. Thus, to avoid further confusion, the authors must carefully choose the terminology and use the names of factors and variables consistently in all the text, including tables, SI and the SAS code. Please, do not make corrections just in places we indicated directly, but revise the entire text. We realize that it is difficult considering complexity of the research presented in this report – but even more the special attention should be given to the consistency of the terminology.

Our response: Yes, thank you. We have now revised the whole text to make it consistent with respect to the naming of factors and variables and consistent use of acronyms.

2. The usage the repeated-measures model with “time” as categorical factor as a bases of the inferences concerning the slope of the freezing extinction

To our comment concerning this issue (Page 5, lines 24-30 of original version) the authors responded:

‘We are somewhat perplexed by this comment. We think that the structure of our repeated measures ANOVA is appropriate. Indeed, the dfs of the nominator of F test was equal 15, which follows from the structure of statistical model. However, to conform to Reviewers’ suggestion we have re-analysed the data with ANCOVA model with the F test with $df=3$ for the line x time interaction. This interaction was still significant ($P=0.015$). Thus, we are re-assured that the opposite trend of fear extinction of the L-BMR mice clearly discernible from animals of other line types (Figure 3) is robust. We therefore elected to retain the repeated measures ANOVA model in the paper. As for possible non-linearity of the fear extinction rate: yes, we tested the significance of the square term. It was not significant.’

The response indicates that the authors have not recognized the merit of our criticism. We never said that the calculation of the dfs for the overall effect of interaction in the repeated measures model was wrong. The repeated measures structure with “time” treated as categorical factor with 6 groups can be indeed applied, but the F test for the overall effect of “time*line type” (with 15 dfs in numerator) is NOT a legitimate basis for the claim that there the line types differ is a systematic trend with time (i.e. that slopes with respect to time differ between the line types). Imagine a simpler case with only 3 time periods, and the mean value of a response variable Y changing with time as follows: 1, 10, 1. The overall F test (with 2 dfs in numerator) would likely indicate a significant effect of “time”, but it would only mean that there are differences among means of the 3 time categories, and NOT that the Y decreases or increases systematically with time; the slope of such a relation is exactly zero. Suppose also that in a second group the pattern of changes in time would be reversed: 10, 1, 10. The overall test of the group*time interaction (with $2*1=2$ dfs in numerator) would probably indicate that the pattern differs between the groups, but this would be NOT a valid bases for a claim that the slopes of the relation between Y and time differ between the groups – because in both groups the slopes equal exactly zero. For the same reason, in the case of your data, the overall test of line type x time interaction with 15 numerator dfs is a valid basis for a claim that there is a heterogeneity in the pattern for the four groups, but NOT a valid bases for the claim that the slopes of the systematic trend in time differ

among the four groups.

In the response the authors say that they have performed (as we recommended) also ANCOVA with time treated as quantitative predictor and got “the same” results – in the sense that the interaction was also significant. However, the fact that in this case conclusions incidentally agreed is not a logically valid basis for reporting results from the non-adequate model. The ANCOVA is still not a perfect solution because it assumes that observations performed for a given individual in the subsequent observations can be treated as independent, rather than representing a time series. Thus, a more elegant and robust, but also more complicated approach, would be to perform analysis of polynomial contrasts within the framework of the repeated measures ANOVA such as you fitted, and test for the interaction between the line-type effect and the first order (linear) component of the five possible orthogonal polynomials. Such an analysis would be more appropriate, because, with a proper residual covariance structure, it would correctly account for non-independence of measurements obtained in the subsequent periods. However, by all appearances, even though the authors specified in the Mixed procedure of SAS the “repeated” statement, in fact they have not fitted a repeated-measures model, because they have specified “variance components” covariance structure (the “type=vc” option in the “repeated” statement). Thus, they actually assumed that the observations at the subsequent time periods can be treated as independent and with equal variances. If preliminary models showed that the assumption is reasonable (i.e., models with more complex covariance structures have not performed better) it also means that the much simpler analysis of covariance with time as the quantitative trait (and ID as random effect) would be also fully legitimate, and much easier to present, because it would directly provide estimates of the desired slopes. Note, that it is still not clear how the parameters of the lines shown on Figure 3 were obtained, because the SAS code you have shown in the supplementary materials certainly has not provided the relevant estimates (actually, the code is shown for a different analysis, but I understand the logic of the model was the same).

Thus, the result of the F test presented on page 6 lines 2-6 (“ $F_{15,30} = 2.27$; $p = 0.03$ ”) would be a valid basis for the statement that it:

“reflected the heterogeneity of the dynamics of fear extinction in the studied line types,” but is NOT a valid basis for the statement that it:

„accounted for the low-BMR mice losing fear response much faster than other line types.”

Again, it is not a valid basis because you could have the significant interaction even if there is no difference in the overall linear trend, i.e., in the slopes of the regression of the response variable on time would not differ.

In addition, in both cases the overall interaction means only heterogeneity among the four groups, and not that a particular group (low-BMR) differs from the others – for the same reason as in any ANOVA with more than two groups the overall test of the grouping effect is not the same as a test that a particular group differs from others. The authors justify the distinction of HBMR group by claiming that they have performed “a priori contrasts”, but this claim rises another concern (next point).

Our response: We have conformed to the Reviewer’s recommendation. We have replaced the repeated measures model with the ANCOVA model, with time as a covariate and the ID as a random effect. As we indicated earlier, this analysis did not change the statistical inference: the line type x time interaction was still significant ($F_{2,530} = 3.34$; $p=0.019$). To address the question of whether the interaction resulted from the differences in slopes of the L-BMR line type vs. other line types we carried out two- custom made tests by means of the Contrast statement, as specified in the SAS code now provided in the Supplementary Materials. In the first Contrast we tested the heterogeneity of slopes of changes in fear response of the RB, H-BMR and PMR line types. The resulting test statistic indicated that the slopes did not differ ($F_{2,530} = 1.51$; $p=0.22$). In the second custom test we asked whether the slope of the L-BMR line type differed from the remaining line types. This difference was statistically significant at $p=0.005$ with $F_{2,530} = 7.79$. Thus, taken together, both custom tests suggest

that the statistical significance of the line type x time interaction in the ANCOVA analysis of fear extinction can be attributed to the difference between the slope of the rate of this extinction in the L-BMR mice vs. the slopes of other line types. However, we anticipate that the Referee can have different opinion and insist on the application of the multiple comparison adjustment. We would therefore like to point out, that even accepting the need for such adjustment (e.g., accepting the alpha value of $0.05/4=0.0125$ with the Bonferroni correction) does not change the outcome of our statistical inference.

We have now provided the description of the ANCOVA in the main text (p. 12, lines 19-21) and the rationale for the use of the contrasts in the Supplementary Materials (lines 102-122).

3. The application of *a priori* contrasts

In response to our comment to Fig. 2 concerning the issue of using “pairwise a-priori t tests” the authors responded:

“We carried out planned custom hypothesis tests by means of the Contrast statement of procedure Mixed. So the correction for multiple tests does not apply here.”

and in Supplementary Information, page 4, lines 102-104 provided a brief information that:

Planned custom hypothesis tests of the differences in slopes of the changes in the number of nose pokes depicted in Figures 2 and 3 of the main text were carried out by means of the Contrast statement of the procedure Mixed of SAS”.

This information was also repeated on Fig. 2 and 3 legends.

The response shows a gross misunderstanding, and in fact deepens the doubt concerning the whole results of the statistical analyses. The “contrast” statement performs a test for a contrast without correction for multiplicity, but whether it is legitimate to apply uncorrected contrasts or not does not depend on what command of the statistical procedure you use, but on whether the contrasts were really a priori planned, and planned with respecting the rules for such contrasts. The limitation for contrasts not requiring the multiplicity correction is that a) the number of such contrasts to the degrees of freedom, i.e., to three contrasts in the case of four groups, and b) that the contrasts are independent (orthogonal). Thus, the analysis of such contrasts, by definition, cannot lead to results that can be presented in the way the authors did it – by using different letters to show means or slopes in four groups differ from each other (such as used on Figs. 1-3).

Our response:

As stated on p. 3, lines 30-31 of the Introduction, we tested for ‘(1) the existence of the brain-gut trade off and (2) positive associations between BMR or PMR and CA and brain size.’ We started our study with testing (1), which identified the H-BMR/L-BMR line types as the ones having the highest/lowest BMRs and largest/smallest internal organs. Therefore, in the behavioural part of our study we concentrated on the comparisons of the H-BMR and L-BMR line types with the remaining line types, by means of the custom made contrasts of the H-BMR or the L-BMR line type vs. the rest. We are therefore confident that this is a legitimate a priori approach. We admit, however, that the letter labelling in Figures 2 and 3 were confusing. We have now deleted them and provided additional information on the contrast use in captions of both figures.

We agree that comparisons of the line type LSMs presented in Figure 1 should be carried out with the correction for multiple comparisons. Indeed, for several pairs of comparisons (e.g., the H-BMR vs. PMR line types) we had no a priori hypothesis in mind. We have therefore re-analysed the respective data with AJUST=BON option, that is, with the Bonferroni correction. It did not affect the statistical inference with respect to BMR. However, it did change it in the case of comparisons of the heart mass (no difference between the PMR and H-BMR line types, other comparisons remained unaffected) and kidneys (all other pairwise differences became insignificant, except for the H-BMR vs. L-BMR comparison). We corrected the letter labelling accordingly, along with the caption of the Figure 1. We have also accordingly corrected the main text (p. 4, line 15) and SAS code provided in the SM.

It is indeed very helpful that the authors showed in SI the SAS code used for the analyses, because it confirmed that the criticism is justified. For the analyses of the traits related to the reward and aversive learning tests, the authors performed 3 contrasts, but the contrasts are not independent (not orthogonal), because in each of the contrast one group is compared with all others.

Our response: We certainly agree that the contrasts specified in the SAS code are not orthogonal. Please note, however, that among those three contrasts we used the only one: the H-BMR line type vs. the other line types, because this was the contrast testing our prediction of the positive association between BMR and CA. We admit that for exploratory analyses, not used in the paper, we have also tested other contrasts. For the sake of clarity we have now deleted them from the SAS code included in the Supplementary Materials.

*Moreover, the code shows also that in fact the authors performed also multiple paired tests between all groups and subgroups without correction for multiplicity (the statements "LSMEANS line/ tdiff", "LSMEANS line*time/ tdiff"). Certainly, it is not legitimate within one set of analyses to perform for the same groups both the test for all possible comparisons among pairs of groups and some selected contrasts without corrections for multiplicity. In fact, in this case the correction for the test concerning the line*time should be made for a total of 41 contrasts: 3 performed with the contrast commands and $(8*8-8)/2=38$ by the "lsmeans line*time/tdiff" command (because in this case "time" is a factor with two groups, so there are $4*2=8$ subgroups). But, obviously, the 38 simple comparisons are not relevant – none provides the answer whether the rate of learning differs between the line types - and it is not clear why they were at all requested.*

Our response. Again, we agree. However, the 'lsmeans line*time/tdiff' command served exploratory purposes, and unlike the Contrast command, was not used in statistical inference. This line has been now deleted from the SAS script.

The SAS code reveals also that uncorrected multiple comparisons were performed also in the analysis for the comparison on mean values of BMR and anatomical traits. Thus, the results on Fig. 1 are not proper, either.

Our response. Letter labelling denoting statistical inference presented in Figure 1 was changed following the application of the Bonferroni correction for multiple comparisons, as detailed above.

Most important, the authors should recognize that they cannot claim that they have performed "planned contrasts" if they never told what the planned contrasts were. The a priori tests are legitimate only if the plan was established before performing the research and seeing the results, and the planned contrasts are compatible with the a priori hypotheses based on external information. Noting in the text of Introduction or Methods indicates that a set of contrasts was a priori planned. The hypotheses presented in Introduction are far too general to be a basis for a valid set of planned contrasts. Note, that the contrasts include all four line types, whereas the random bred (RB) line type is not even mentioned in Introduction. Moreover, even if you could present a specific a priori hypothesis for the comparison of the High-BMR vs Low-BMR lines, nothing in the text of Introduction indicates that you had such hypotheses for the other possible comparisons. Actually, even in the response to our comments you made it clear that only for this comparison you had a strong basis, whereas the other lines were of secondary importance and their inclusion on the analyses had an exploratory character. Therefore, applying the a priori contrasts (even if they would be technically correctly specified) was not methodologically valid, and consequently the reported „significance" is not justified - not only for results presented on

Fig. 2 and 3, but also for those on Fig. 1.

Our response. As explained above, we had a strong basis for a specific hypotheses related to the H-BMR and L-BMR line types. To make it clear we have now added a relevant explanation in the Supplementary Materials (lines 102-122). Please also note that from the very beginning, the between line type differences in the LTP presented in Figure 4 were tested by means of the Tukey post-hoc test, because in this case we had no clear a priori predictions.

Obviously, if after all my comments the authors would now present a clear set of hypotheses leading to a set of orthogonal contrasts, those would be not “a priori” hypotheses and contrasts. Therefore, in my opinion the only proper solution is to perform the post-hoc comparisons, with appropriate correction for the multiplicity of a given set of tests. Note, that even if the results would force changing some conclusions, i.e., some effects would no longer appear significant, this would not decrease the value of the work. We all agree that “p-hacking” is not the path we want to follow.

Our response. We disagree. We did have a set of clear ‘a priori’ hypotheses, as formulated in the contrasts statements already applied in the analyses presented in the first version of the paper. We therefore see no reason to correct them for multiple comparisons. However, we are confident that the way of our presentation of the statistical tests will allow any unconvinced reader to apply any correction of his/her choice. As we showed above, the application of the Bonferroni correction (which is widely considered most conservative) does not affect the inference with respect to the use of Contrast statements.

4. The logic of the repeated measures model for the results of the reward and aversive learning

*According to the description in text (page 10, lines 18-23) the data were automatically summed for 12-hour continuous periods, but the results are reported for the critical 24 h periods (the last day with water and the first day with sugar or quinine). The SAS code in SI shows that actually the response variable was the number of nose pokes in one of the “phases”, but the “phase” refers to photoperiod phase (light or dark) – and the correspondence between continuous 12 h periods and the light/dark phases would be strict only if the change of the state from “water” to “sugar” or “quinine” was made at the onset of light phase or onset of dark phase. Otherwise, one of the phases would actually consist of two parts, split by the other photoperiod phase. The authors do not inform at what hour the bottles were changed, so the reader cannot guess. Splitting the timing would not affect the means for the whole 24h periods, but it would certainly affect the estimate of the effect of the “phase”. This was not a focal aspect, but the problem affects also validity of the specification of the repeated measures model, in which the “repeated” factor was encoded as the “time*phase” interaction (where “time” variable is used, inconsistently with declaration in Methods, as the categorical factor distinguishing two “experimental phases” or Days). The repeated measures factor should describe the real order of obtaining the results. If the timing of the two 24 h periods is not aligned with the light/dark phase, the coding used by the authors does not describe properly the structure of the repeated measures design. In addition, even if it is aligned, but the bottles were replaced at the onset of light phase, the repeated statement used by the authors would not provide the proper order, because by default SAS orders the groups defined by text codes alphabetically, and therefore phase=dark comes before phase=light. Thus, the order of day/night would be reversed (this could be amended by changing the codes of the categories, or requesting a change of reference group). The coding of the repeated measures design would be correct only in the case when bottles were changed at the onset of dark phase. If the bottles were, as I suspect, changed sometime during the day, all the repeated measures model is invalid.*

Note that, even though the results of the significance tests for the “phase” factor and

*“time*phase” interaction are reported in Table 2 (but named there “Period” and “Day”), they are not mentioned at all in Results or Discussion; all the inferences are based only on the test for the effects of “time” (Day) and “line” (which in the entire test is named “line type”, whereas “subline” in SAS code corresponds to “line” in all the other text...). Also, all the least-squares means you show on figures are values averaged across the two 12 h periods, i.e., represent whole days. Thus, why you have at all bothered with performing and presenting the complicated analysis with the repeated-measures model for 12 h periods, rather than a much simpler model in which the dependent variable is a value averaged across whole 24 h periods? Yes, the effect of phase/period was significant, but this is really a trivial observation that the behavior of mice differs between light and dark phase. Perhaps more interesting are the significant interactions between the phase and line type – but you have not discussed the results, either. Note, that with the simpler model Table 2 would have 3 columns less, and the readers would be not distracted by many F and p values that are never referred to in the text. I do not insist you should make this radical change, but if the switch of light/dark phases is not synchronized with the timing of the switch from water to sucrose or quinine, this would probably be the most straightforward solution to the difficulty of specifying a correct structure of the repeated measures model. If the authors would like to discuss the above statistical issues, please contact pawel.koteja@uj.edu.pl (one the persons who prepared this review).*

Our response: The Referee is correct. The sequence of 12 h light and dark photoperiod phases is important and of course, has been taken into account in experimental design and subsequent analyses of the results. We started behavioural tests with the dark Period (earlier referred to as dark ‘phase’) and switched the bottles at the onset of the second dark Period. Thus, the sequence of Periods coded in the statistical analysis did reflect the experimental design. We have now re-written the description of the ANCOVA model (p. 12 line 3 of the main text) to add this information.

Minor points

Page 2 lines 6-8 - *We fully understand your decision to use the terms “line type” as used in Garland's group. Note, however, that unselected, random bred lines are not mentioned – neither here nor in the Abstract in general. Perhaps you could add the phrase “, and unselected lines” at the end of this sentence and save three words in other places of Abstract. For example, in lines 11/12 the phrase “to the other line types” is not necessary, because the previous sentence makes it clear to what the superiority refers.*

Our response: We have rewritten parts of the Abstract to incorporate the phrase ‘unselected lines’. However, this incurred other minor changes (highlighted in green in a marked copy of the paper), which were necessary to conform to the 200 words limit.

Page 3 line 25 - *Again, unselected, random bred line type is not mentioned. Therefore, the first sentence of Results section (page 4. line 9) is confusing, because it speaks about four line types, whereas only three are mentioned in Introduction.*

Note, that the exemplary sentence we provided in our previous review included the unselected lines, but you have omitted this part.

Our response: Mentioning unselected lines in this context would be confusing, because the remaining part of the sentence refers to ‘traits widely accepted as pre-requisites for the evolution of homeothermy and large brain size’. Clearly, unselected mice are not relevant in this context. However, we have re-worded the first sentence of the Results section to introduce the acronyms of all line types.

Page 3 line 31, page 7 line 7 - *PMR, CA and LTP acronyms were introduced or spelled out*

earlier in the main text (even though CA and LTP were introduced in the abstract, it should be explained in the main text, too)).

Our response: Done.

Page 4 line 15 - the acronym H-BMR was spelled out (High-BMR) earlier in the text, but LBMR was not. However intuitive the annotation may seem to be, such acronyms should be introduced properly. Perhaps even as early as in page 3 line 25, where the line types are mentioned for the first time.

Our response: All acronyms are now spelled out in the first sentence of the Results section.

Page 4 line 22 - I believe your conclusion was meant to state that you did not observe the tradeoff and did not find a correlation between brain size and BMR. However, the phrasing of the second part of the sentence, starting with "as well as showed", makes an impression that you did find the brain-BMR correlation. I suggest to rephrase the statement to make the point clearer. Also, by making an emphasis on the personal aspect ("we did not observe, [we] showed") you actually decrease the strength of the results. I think that a simple statement would serve better, e.g.: "... genetic drift showed neither brain-gut trade-off nor positive genetic correlation between BMR and brain size."

Our response: Indeed, wording suggested by the Referee is much better. Corrected accordingly.

Page 5 line 3 and throughout the text - there is an inconsistency in the BMR-selected line type names: sometimes the "High" or "Low" is capitalized and sometimes not, and occasionally it is abbreviated, and sometimes written without the dash between the "high" and "BMR". Note, that we have made the same comment in first review, but some of the inconsistencies remained not amended.

Our response: Corrected.

Page 5 lines 24 and 29 - these two statements are contradictory: "when sucrose solution was replaced with quinine" and "once the bottle with tap water was replaced with the bottle containing quinine solution". We pointed out the issue of "replacing" in our previous review, and the revision amended it only partly.

Our response: Corrected.

Page 5 line 34 - page 6 line 2 - this sentence is confusing for a first-time reader. Literarily, you wrote that after conditioning phase, you measured the extinction of freezing... "in another group of mice", i.e. not those that were conditioned. The information that another group of mice was used would be better given in the previous sentence (Finally, in another group..., we investigated...). Also, even though this is obvious to specialists, you could say more explicitly that response to "perceived threat" is investigated with no shocks applied ("...response to perceived threat (i.e., in the absence of the electric shocks)").

Our response: Corrected.

Page 7 lines 16-17 - the text is not grammatically or logically correct. The rhetoric "both" requires "and" linking some two objects or statements. It could be, e.g., "Both, learning the position of bottle with sucrose and quinine in the tests performed in the IntelliCage, involve hippocampus." However, as this is Discussion, perhaps it would be better to make a more

general statement, such as "Both aversive and positive learning, such as in the tests we performed in IntelliCages, involve hippocampus" (but this is surely up to the authors).

Our response: This is an excellent suggestion. Done.

Page 7 line 28 - *the abbreviated label of the random-bred line type (RB) was not introduced in the text. Again, I suggest introducing all line types and relevant abbreviations already in the Introduction, which will help to use the abbreviations consistently.*

Our response: All acronyms have been introduced in the first line of the Results section.

Page 8 lines 10-11 and 17 - *misleading mental shortcut: divergent selection is applied to mouse lines, not to mice. Saying that mice were selected suggests that animals expressing a particular trait were picked from a general population for the purpose of a given test (as in "we selected patients with the highest / lowest heart rate for a cardiac drug study"). Selection applied on lines implies multigenerational, evolutionary design. We realize that the shortcut form is fully understandable among researchers familiar with experimental evolution, but our own experience shows that many physiologists and neurobiologists from biomedical community have a problem with proper understanding the selection experiments, and therefore it is better to be explicit. Also, the phrase "for high/low BMR" literarily means that you selected for a ratio of high to low BMR. The sentence should rewrite as: "we maintain lines of mice divergently selected for high or low body-mass-corrected Basal Metabolic Rate (BMR), quantified..."*

Our response: Corrected.

Page 9 line 17/18 – *"transponder implantation procedure."*

Our response: Corrected.

Page 9 line 22 - *the statement regarding "excluding" competition for access to the bottles seems to be too strong. There were less bottles than there were mice, and during the learning and test phase each mouse had to share access to their only source of water (a single cage corner) with one or two other mice. In such situation competition cannot be fully excluded, but it can be reduced/limited/minimized, as you phrase it on page 10 line 2.*

Our response: Corrected.

Page 9 line 29 - *"pokes" should be in past tense.*

Our response: Corrected.

Page 10, line 11 - *"During the next phase lasting 2 days," – either comma after „phase" (as in line 13), or in this case better: „During the next 2-day phase".*

Our response: Corrected.

Page 10 lines 13-14 - *I do not understand "the bottle preferred during the previous two days". During the previous two days both bottles contained tap water. It can be imagined that a mouse may prefer to drink water from one of the two bottles it is offered, but the or two other mice assigned to the same corner (page 10 line 1) can vary in their spatial preferences. Hence, placing*

quinine solution in place of a water bottle preferred by one of the mice would likely mean replacing the less-preferred water bottle of another mouse.

Our response: Yes, in theory, replacing bottles may result in the confounding effect highlighted by the Referee. In practice, however, the majority of mice shared spatial preferences and use the same bottle, so the confounding effect is negligible.

Page 12 line 15 - "drawn", not "drown"

Our response: Corrected.

Page 17, Figure 1 legend - repeated information in lines 8-9 and 13-15. In line 9, before "RB-" there should be ", and" rather than period.

Our response: Figure caption has been re-arranged to avoid repetition.

Figure 1 – to our comment on previous version, in which we pointed out that the bar graphs should begin with zero on Y axis, the authors responded:

"we disagree. If the differences are exaggerated, then the respective SEs are exaggerated too, so visual proportions between differences and errors of estimate remain unaffected."

The response is not adequate. Indeed, the proportion of the difference to error is represented correctly, but in the case of bar graphs our brains tend to first of all spot the ratios of the heights of the bars for different groups. Thus, when we look at Fig. 1a, the first thing we notice is that the H-BMR bar is twice as high as the L-BMR one, and only after reading the values on the Y axis we can realize that the gap between 40 and 80 indicates that the first assumption cannot be true - but it is impossible to tell by how much, because there is no label at the intersection of the x and y axes. This is not only an inefficient, but also a misleading way of presenting the data. The rule that bar charts should start from zero is widely recognized by experts of scientific communication. If you do not believe, just google "bar chart starts zero", and in front of the long list of resources you can get, e.g.:

<http://www.chadskelton.com/2018/06/bar-charts-should-always-start-at-zero.html>

where the first sentence says: "If there's one thing almost everyone agrees on in data visualization, it's that bar charts should start at zero." As the authors are from Poland, you can consult also the guide of January Weiner "Technika pisania i prezentowania...", which some of the authors surely know.

Our response: We would truly like to avoid a protracted discussion with the Referee on the legitimacy of the use of truncated Y axes in our Figure 1 bar chart. We have therefore changed the style of the Figure 1, albeit retaining truncated Y axis. It echoes the style of Figure 2a presented in the paper recently co-authored by the Referee (Lipowska et al., Sci. Reports, 2022). We therefore hope that our new Figure 2 meets the Referee's expectations. This figure style has also been recently used in a paper published in the RSPB (Figure 2b, Royles et al., 2021), so our understanding is that it also meets the RSPB standards.

Figure 1a - Y axis unit is "ml zero-squared per hour"?

Our response: Of course, O₂. Corrected.

Fig. 1b brain - I understand that there were no differences in brain mass among types of selection. Why bother putting all the letters "abcd"? Why not just "a" above each?

Our response: Yes, thank you, corrected.

Figure 2 description, line 2 and 6 - "row" not "raw".

Our response: Corrected.

Figure 2 legend line 13 and 15 – "least square means" - it should be "squares".

Our response: Corrected.

Figure 2 - again, there is no reason to use multiple letters "cd" on Fig. 2b and "bcd" of Fig. 2c. More importantly, to our suggestion that Y axes on Figs. 2Ba 2C should have the same scale, the authors responded that:

"The tests are not directly comparable, as they address different aspects of learning. We would therefore like to retain different scales."

In our opinion the argument is not satisfactory. Even though different aspect of learning is assessed, the same behavior is directly measured, and in both tests the starting condition is the same ("water"). Therefore, it is meaningful to ask whether the behavior of the animals at the start was the same. The different scales on the Y axes obscure the fact that the number of nosepokes with pure water was only about 15-20 in the first test, but about 25-38 in the second test. The difference was in fact of the same order as that between water and sugar. The authors have not tried to explain the difference, which – with the figure properly prepared – would be more striking and would certainly rise a question about the explanation. One possible explanation is that the test was performed on a different group of animals. However, if we correctly understand the description of the scheme, the results presented on Fig. 2B (test with sugar) are pooled results from the two groups of animals – correct? If this is the case, did the two groups differ in the first test, and if yes, what could be the reason? Another possibility is that passing the first test modifies behavior of the mice so that they behave in a different way also under exactly the same condition (in the phase with water). If this is the case, however, the results of the second test – of the aversive learning – cannot be treated as independent of the first one. In other words, can we be sure that the results of the aversive learning would be the same if the test were performed on really naïve animals, rather than those that had passed through the reward learning test? Thus, we suggest to not only amend the presentation (show the same scale on Y axes), but also to discuss shortly both the difference in the results with water, and the issue of non-independence of the two learning tests.

Our response: We respectfully disagree. As explained on p. 5, in lines 18-22 the rationale for using two different paradigms was as follows: 'To exclude a possibility that the superior learning response of the H-BMR mice was solely limited to the reward-seeking context or higher motivation to perform a nosepoke response, we used another group of naïve mice and carried out a study designed in the IntelliCage system as described above, but with the reward-seeking discrimination learning followed by an aversive cue discrimination task (Figure 2A).' We therefore did our best to differentiate the results of both tests, and this included their graphic presentation, with the use of different Y scales.

We also think that it is clear from the above quote, that for the aversive-cue task we used a separate group of naïve mice. It was therefore reasonable to test whether this new group of animals responded to the reward-seeking context in a repeatable manner consistent with that, found in the group tested in a previous reward-seeking trial represented in Figure 2B. Indeed, both groups of mice responded to the reward-seeking context consistently, as reported on p. 5, in line 24-26 of the main text. This assured us, that the outcome of the aversive cue discrimination task was not affected by possible inconsistency in the response to the reward-seeking context. Both parts of the trial were

separated by a 2 days phase with all bottles filled with water (being a neutral cue). Subsequently, the preferred bottle was replaced with the one containing quinine. We therefore think that it is highly unlikely that the outcome of part of the trial with the aversive cue was confounded by its reward-seeking component.

Figure 3 legend (and methods regarding this measurement) - still not sure what was measured, what does the percent represent. Was the behavior scored second by second, and a fraction of immobile seconds within each 30s period was calculated per mouse? Or were the mice given a 0 or 1 score depending on whether a particular 30-s interval featured at least one incident of freezing, the percent representing a line mean? This should be clarified better in the text.

Our response: Freezing was defined as the cessation of all movements, except for respiratory ones, for at least 1 s. The cumulative time spent freezing was converted to a percentage score in each of the 30-s intervals for each individual mouse. We have clarified this on p. 11, lines 7-10 of the main text.

Table 2 line 6 - this is the first place in the text where the word "batch" appears. The effect was not mentioned in the Results or Methods sections. In the SI file in line 98 the word appears, as one of the effects included in model, but with no explanation of what the "batch" means. We can only guess that "batch" distinguish between the animals that were used in only the reward learning and those that passed both tests. The significant effect of batch can provide a clue concerning my earlier inquiry concerning the difference in number of nose pokes during the phase with pure water (Fig. 2).

Our response: As we stated on p. 24, line 6 of the main text by 'batch' we mean 'a group of animals simultaneously subjected to behavioral test'. For clarification, we have now repeated this definition also in lines 98-99 of Supplementary Materials.

We had only 3 IntelliCage setups (p. 9 line 26 of the main text) at our disposal, so it was impossible to carry out behavioural tests on all animals at once. For this reason, we controlled for the effect of 'batch' in our statistical analyses.

Suppl., line 1 - missed one "p" in Supplementary."

Our response: Corrected.

Suppl., line 42 – even though LTP is explained in the main text, it would be good to be explained also in the first occurrence in the Supplement.

Our response: Done.

Suppl., line 109 - what was randomly selected - animals, replicate line, or both? It can be said better in a form of: " We have therefore restricted these measurements to mice from only three lines: H-BMR, LBMR and one randomly selected replicate line belonging to the RB line type, which was treated as a reference."

Our response: Thank you, we have reworded the sentence accordingly.

Figure S1 - the blue line of the confidence interval looks like just another plotted variable, which for some reason was not introduced in the figure legend. The distinction would be clearer if the line was drawn in a different style, e.g. without symbols.

Our response: Yes, corrected.

Table S1 - I appreciate inclusion of such a detailed table, but one fragment is puzzling. According to Methods section the sucrose preference test was performed on two groups of mice, with the test on the second group extended by a quinine avoidance test. However, the generation numbers are completely different for the rows concerning the test with sucrose and quinine. Thus, this would suggest that, even though on the second group the test with sucrose was performed, the results were not used in the data analyses. Is that correct? But if this is the case, what is the “batch” effect in the ANCOVA model? Also, should not the row for quinine be labelled “incorrect” (as in other tables)? An extra column presenting a total number of animals in each group would also be appreciated, even if it is not entirely necessary for proper comprehension.

Our response: Yes, thank you, the column heading should read ‘Incorrect’.

As explained above, the aversive and positive learning behavioural tests were carried out on separate groups of naïve animals. Yes, they belonged to different generations, as indicated in Table S1. As also explained above, the ‘batches’ were subgroups of animals ‘simultaneously subjected to behavioral test’.

The total numbers of animals from each line type used for particular analyses are reported under the heading ‘n’ in the respective columns of Table S1.